# The Effect of Acute Oral Galactose Administration on the Redox System of the Rat Small Intestine

**DOI:** 10.3390/antiox11010037

**Published:** 2021-12-24

**Authors:** Jan Homolak, Ana Babic Perhoc, Ana Knezovic, Jelena Osmanovic Barilar, Davor Virag, Mihovil Joja, Melita Salkovic-Petrisic

**Affiliations:** 1Department of Pharmacology, University of Zagreb School of Medicine, 10 000 Zagreb, Croatia; ana.babic@mef.hr (A.B.P.); ana.knezovic@mef.hr (A.K.); josmanov@mef.hr (J.O.B.); davor.virag@mef.hr (D.V.); mjoja@student.mef.hr (M.J.); melitas@mef.hr (M.S.-P.); 2Croatian Institute for Brain Research, University of Zagreb School of Medicine, 10 000 Zagreb, Croatia

**Keywords:** galactose, oxidative stress, gastrointestinal tract, redox, redox homeostasis

## Abstract

Galactose is a ubiquitous monosaccharide with important yet incompletely understood nutritive and physiological roles. Chronic parenteral d-galactose administration is used for modeling aging-related pathophysiological processes in rodents due to its ability to induce oxidative stress (OS). Conversely, chronic oral d-galactose administration prevents and alleviates cognitive decline in a rat model of sporadic Alzheimer’s disease, indicating that galactose may exert beneficial health effects by acting in the gut. The present aim was to explore the acute time-response of intestinal redox homeostasis following oral administration of d-galactose. Male Wistar rats were euthanized at baseline (*n* = 6), 30 (*n* = 6), 60 (*n* = 6), and 120 (*n* = 6) minutes following orogastric administration of d-galactose (200 mg/kg). The overall reductive capacity, lipid peroxidation, the concentration of low-molecular-weight thiols (LMWT) and protein sulfhydryls (SH), the activity of Mn and Cu/Zn superoxide dismutases (SOD), reduced and oxidized fractions of nicotinamide adenine dinucleotide phosphates (NADPH/NADP), and the hydrogen peroxide dissociation rate were analyzed in duodenum and ileum. Acute oral administration of d-galactose increased the activity of SODs and decreased intestinal lipid peroxidation and nucleophilic substrates (LMWT, SH, NADPH), indicating activation of peroxidative damage defense pathways. The redox system of the small intestine can acutely tolerate even high luminal concentrations of galactose (0.55 M), and oral galactose treatment is associated with a reduction rather than the increment of the intestinal OS. The ability of oral d-galactose to modulate intestinal OS should be further explored in the context of intestinal barrier maintenance, and beneficial cognitive effects associated with long-term administration of low doses of d-galactose.

## 1. Introduction

Galactose is an omnipresent epimer of glucose that was first described by Louis Pasteur in 1856 following Erdmann’s observation that hydrolysis of the milk sugar lactose yields a substance that is not glucose. Both in its free form and attached to other molecules forming oligo- or polysaccharides, glycolipids, or glycoproteins, galactose is ubiquitous in all living organisms [1]. Furthermore, together with glucose (in the form of disaccharide lactose) galactose is a cornerstone of animal milk that provides structural and metabolic support during the most sensitive developmental period [1]. Regardless of its importance as a nutrient and pertinent physiological role best reflected in biological consequences of inherited defects of its metabolism, the biochemistry of galactose and its implications in health and disease remain enigmatic [2,3,4].

Galactose is extensively used for modeling aging-related pathophysiological processes in rodents [5,6,7,8]. Oxidative stress (OS) has been proposed as the main driver mediating galactose-induced senescence, although exact pathophysiological mechanisms mediating detrimental effects are yet to be elucidated [5,6,7,8]. Proposed mechanisms by which galactose may induce redox dyshomeostasis include: (i) increased leakage of electrons from the mitochondrial electron transport chain [9], (ii) wasting reducing equivalents (nicotinamide adenine dinucleotide phosphate; NADPH) that are required for biosynthetic and redox reactions in the process of metabolism of galactose to galactitol catalyzed by aldehyde reductase (AR; EC 1.1.1.21) [5], (iii) osmotic stress induced by accumulation of galactitol [10], (iv) generation of advanced glycation end products (AGE) [5,6], and (v) accumulation of H_2_O_2_–A byproduct of galactose oxidation catalyzed by galactose oxidase (GO; EC 1.1.3.9) [5,6,11]. Although a large body of evidence supports the ability of exogenous administration of galactose to induce OS in rodents [5,6,8], serious methodological challenges stand in the way of understanding the biological relationship of galactose and redox homeostasis: (i) the majority of galactose-related animal studies exploited the effects of repeated administration of galactose for induction of ageing-related pathology and were not designed to provide an insight into the biochemical fate of galactose and its implications for health and disease; (ii) studies on neurobehavioral and OS-related alterations in galactose-induced rodent models of ageing are of relatively low quality, high heterogeneity, and are considered to be highly susceptible to bias (e.g., important measures to reduce the risk of bias were reported only rarely) [8]; (iii) the effects of dose/time-response, route of administration, region-dependent metabolic capacities, and the presence/absence of underlying pathophysiological processes are still insufficiently explored, and (iv) the observed association of galactose administration and OS is seldom questioned, with little concern for the lack of mechanistic explanations. The latter is well illustrated by the fact that the GO-catalyzed generation of H_2_O_2_ is often referred to as a probable mechanism of galactose-induced OS and senescence in rodents [5,6,11] regardless of the fact that this is a fungal enzyme that does not exist in rodents [3,12].

In contrast to accumulated reports of its detrimental effects, a substantial body of evidence speaks in favor of the physiological importance and health-promoting potential of galactose. Glycans, long-branched oligo- or polysaccharides covalently attached to macromolecules forming glycoconjugates are increasingly recognized for their high-density information-coding capacity and critical importance in most biological functions [4]. Galactose is an essential substrate in glycan biosynthesis and one of the most abundant monosaccharides in the “sugar code” [4]. Galactose can be utilized in glycosylation via multiple biochemical routes [13], and the presence of even a low concentration of galactose enables the cells to maintain mature glycosylation patterns and growth factor signaling in the conditions of sugar starvation (an effect not observed for glucose or mannose) [14]. Furthermore, galactose can be utilized for energy storage and production, the replenishment of antioxidants, and synthesis of nucleotides and nucleic acids depending on tissue demands with NADPH generating pathways favored over energy metabolism [1,3,4]. Taken together, a unique biochemical fate of galactose hints that it might have been evolutionarily selected as an ubiquitary sugar in the living world and a component of lactose for its ability to complement glucose (primarily used for energy metabolism) by first supplying the anabolic pathways (maintaining glycosylation and fueling reductive biosynthesis) and then serving as a secondary substrate for energy production.

The health-promoting effects of galactose have been demonstrated in a rat model of sporadic Alzheimer’s disease (sAD), in which chronic oral galactose administration was able to both prevent [15] and restore [16] cognitive deficits induced by intracerebroventricular administration of streptozotocin (STZ-icv). Although the main mechanism by which galactose counteracts the effects of STZ-icv still remains unresolved, replenishment of intracellular glucose in a streptozotocin-induced insulin-resistant brain state [15], secretion of the neuroprotective incretin glucagon-like peptide-1 (GLP-1) [16], restoration of brain energy metabolism [16,17], and regeneration of brain NADPH by disinhibition of the oxidative pentose phosphate pathway flux [3] have all been reported and proposed as possible explanations. Importantly, in contrast to the majority of studies utilizing galactose for the induction of aging-related pathophysiological processes in which galactose was standardly administered via the parenteral route, health-promoting effects of galactose in the rat model of sAD have been repeatedly observed following long-term administration of galactose dissolved in drinking water (200 mg/kg/day) and available *ad libitum* [15,16], indicating that (i) the gastrointestinal tract might be a critical factor mediating the observed beneficial effects, and (ii) galactose might be beneficial only in small “tolerable” doses when tissue galactose metabolic capacity is not saturated. The latter is supported by the findings that both oral and subcutaneous administration of d-galactose may exert positive effects on learning and memory [18], and that repeated administration of even relatively low concentrations of d-galactose (100 mg/kg) can induce cognitive impairment, OS, and alter the activity of mitochondrial respiratory chain complexes when administered by repeated bolus dosing [19,20]. The possibility that the gastrointestinal tract mediates the protective effects of galactose is especially interesting given the increasingly recognized importance of gut homeostasis in the context of etiopathogenesis of neurodegeneration [21]. As different pathophysiological alterations of the gastrointestinal tract have been recently reported in the STZ-icv rat of sAD (e.g., redox dyshomeostasis [22], dysfunctional cell turnover and suppressed apoptosis of the intestinal epithelium [23], altered function of the brain–gut GLP-1 [22] and glucose-dependent insulinotropic polypeptide (GIP) [23] axes), it is possible that oral d-galactose treatment is able to exert neuroprotective effects by alleviating intestinal dyshomeostasis.

Considering the relevance of gut redox homeostasis for the maintenance of intestinal barrier and function [24,25] and a unique biochemical role of galactose in the context of redox regulation [3], the primary aim was to explore whether acute oral administration of d-galactose can “oversaturate” metabolic and redox capacity of the small intestine and induce OS. Our secondary aim was to explore the effects on individual redox-related systems and their relationships to better understand the biochemical background of the effects of d-galactose on redox homeostasis.

We hypothesized that acute oral administration of d-galactose would not be able to induce redox dyshomeostasis that would result in OS in the rat small intestine.

## 2. Materials and Methods

### 2.1. Animals and Experimental Design

Three-month-old male Wistar rats from the animal facility of the Department of Pharmacology (University of Zagreb School of Medicine, Zagreb, Croatia) were used in the experiment. The animals were kept 2–3/cage in the controlled environment with a 12 h light/dark cycle (7/19CET), 21–23 °C temperature, and 40–70% relative humidity. Standard bedding (Mucedola S.R.L., Italy) was changed twice per week. Standardized food pellets and water were available *ad libitum*. Rats were randomly assigned to 4 groups (*n* = 6/group). One group received no treatment (baseline control [CTR; *n* = 6]), while all other animals were administered 1 mL of galactose solution (200 mg/kg dissolved in H_2_O (0.55 M)) via a flexible oral gavage feeding tube. Oral galactose-treated animals were euthanized 30 (30; *n* = 6), 60 (60, *n* = 6), or 120 (120, *n* = 6) minutes following galactose administration (Figure 1). The treatment dose and time-points were chosen based on our previous work [3,15,16,17] and represent a human equivalent dose [26] of 32.4 mg/kg (~2 g dose in a 60 kg human) corresponding to approximate daily galactose intake [27] (not including lactose-derived galactose).

### 2.2. Tissue Collection and Sample Preparation

Animals were euthanized in general anesthesia (ip ketamine 70 mg/kg, xylazine 7 mg/kg) and decapitated. The post-gastric 2 cm of the proximal duodenum and pre-caecal 2 cm of distal ileum were dissected out and luminal contents were rinsed using a syringe with ice-cold phosphate-buffered saline (PBS). Intestinal segments were then cut open, rinsed in ice-cold PBS, snap-frozen in liquid nitrogen, and stored at −80 °C. The samples were homogenized on dry ice using mortar and pestle, dissolved in 500 μL of lysis buffer (150 mM NaCl, 50 mM Tris-HCl pH 7.4, 1 mM EDTA, 1% Triton X-100, 1% sodium deoxycholate, 0.1% SDS, 1 mM PMSF, protease inhibitor cocktail (Sigma-Aldrich, Burlington, MA, USA) and phosphatase inhibitor (PhosSTOP, Roche, Basel, Switzerland) (pH 7.5)) on ice and homogenized using an ultrasonic homogenizer (Microson Ultrasonic Cell 167 Disruptor XL, Misonix, Farmingdale, NY, SAD). Homogenates were centrifuged for 10 min at 4 °C using a relative centrifugal force of 12,879× *g* and supernatants were stored at −80 °C. Out of 48 biological specimens (24 duodenal and 24 ileal samples), 4 were excluded from all biochemical analyses as they were accidentally stored for 48 h at −20 °C instead of −80 °C before protein measurements. The samples were not excluded post hoc, but before the analyses out of precaution to exclude possible bias that might have been introduced by mishandling. The samples that had to be excluded were duodenal sample 3 from the group [30], and sample 6 from the group [120], and ileal sample 1 from the group [0], and sample 2 from the group [60] so in total there were 22 duodenal samples (*n*_0_ = 6; *n*_30_ = 5; *n*_60_ = 6; *n*_120_ = 5) and 22 ileal samples (*n*_0_ = 5; *n*_30_ = 6; *n*_60_ = 5; *n*_120_ = 6) analyzed in the study.

### 2.3. Protein Content Determination

Protein content was measured using the Bradford protocol adhering to the manufacturer’s instructions. Briefly, 5 μL of each homogenate was dissolved in 195 μL of PBS, 30 μL of the Bradford reagent (Sigma-Aldrich, USA) was added to 1 μL of the diluted sample, and a change in absorbance at 595 nm was quantified using the Infinite F200 PRO multimodal microplate reader (Tecan, Männedorf, Switzerland). The protein concentration was assessed with a linear model using a dilution curve of serial concentrations of bovine serum albumin dissolved in lysis buffer.

### 2.4. Lipid Peroxidation

Lipid peroxidation was assessed based on quantification of lipid peroxidation end products using the thiobarbituric acid reactive substances (TBARS) assay as described previously [3,22]. Supernatants of tissue homogenates (12 μL) were mixed with 120 μL of the TBA-TCA reagent (0.038% thiobarbituric acid (Kemika, Croatia) in 15% trichloroacetic acid (Merck, Kenilworth, NJ, USA). The samples were diluted with 70 μL of ddH_2_O. Samples were vortexed, placed in perforated microcentrifuge tubes, and incubated for 20 min in a heating block at 95 °C. The colored adduct of thiobarbituric acid and TBARS (e.g., malondialdehyde (MDA)) was extracted in 220 μL n-butanol (T.T.T., Sveta Nedelja, Croatia) and the absorbance of the butanol fraction was measured at 540 nm in a 384-well plate using the Infinite F200 PRO multimodal plate reader (Tecan, Switzerland). The MDA concentration was estimated using a standard dilution curve of MDA tetrabutylammonium salt (Sigma-Aldrich, USA). Plasma TBARS raw data from Homolak et al. (2021) [3] was used to control whether the observed alterations of redox parameters (including intestinal TBARS) were reflecting systemic or local tissue redox homeostasis.

### 2.5. Nitrocellulose Redox Permanganometry

Nitrocellulose redox permanganometry (NRP) was utilized for the determination of tissue reductive capacity [3,22,28,29]. A measure of 1 μL of each homogenate was loaded onto the nitrocellulose membrane (Amersham Protran 0.45; GE Healthcare Life Sciences, Chicago, IL, USA) and left to dry out at room temperature. The dry membrane was incubated in the NRP reaction solution (0.2 g KMnO_4_ in 20 mL ddH_2_O) for 30 s and washed in dH_2_O until only the MnO_2_ precipitate was trapped in the nitrocellulose. Membranes were digitalized and analyzed in Fiji (NIH, Bethesda, MD, USA) [28].

### 2.6. The Concentration of Low-Molecular-Weight Thiols and Protein Sulfhydryl Content

Low-molecular-weight thiols (LMWT) and protein sulfhydryl content (SH) were measured by quantification of 5-thio-2-nitrobenzoic acid (TNB) in the reaction between reactive sulfhydryl groups present in the supernatant and protein precipitate of the tissue sample and 5,5′-dithio-bis(2-nitrobenzoic acid) (DTNB) [3,30]. A measure of 25 μL of each homogenate was incubated with the same volume of sulfosalicylic acid (4% w/v) for 1 h on ice. The samples were centrifuged for 10 min at 10,000 RPM and 30 μL of the supernatant was transferred to separate wells. Both the transferred supernatant (used for estimation of LMWT) and the protein pellet (used for estimation of SH) were mixed with 35 μL of the DTNB reagent (4 mg/mL in 5% sodium citrate) and left to react for 10 min. The absorbance of the supernatant from both reactions was read at 405 nm using the Infinite F200 PRO multimodal microplate reader (Tecan, Switzerland). Both SH and LWMT concentration was calculated using a molar extinction coefficient of 14,150 M^−1^cm^−1^.

### 2.7. Superoxide Dismutase Activity

Superoxide dismutase (SOD) activity was determined based on the inhibition of the 1,2,3-trihydroxybenzene (THB) autooxidation rate [31,32]. A measure of 3 μL of each sample was incubated with 100 μL of the reaction buffer containing 80 μL of the THB solution (60 mM THB dissolved in 1 mM HCl) added to 4000 μL of the SOD reaction buffer (0.05 M Tris-HCl, and 1 mM Na_2_EDTA (pH 8.2)) or MnSOD reaction buffer (0.05 M Tris-HCl, 1 mM Na_2_EDTA, 2 mM KCN (pH 8.2)) right before incubation. A measure of 2 mM KCN was used for the discrimination of Cu/Zn-SOD from Mn-SOD [33,34,35]. The increment of absorbance at 450 nm (indicating THB autooxidation) was measured using the Infinite F200 PRO multimodal microplate reader (Tecan, Switzerland) with kinetic interval time cycles of 30 s.

### 2.8. Catalase Activity

Catalase (CAT) activity was determined by measuring the dissociation rate of H_2_O_2_ using the adaptation of the method proposed by Hadwan [36]. A measure of 18 μL of each homogenate was incubated with 40 μL of 10 mM H_2_O_2_ in 1 × PBS and the reaction was stopped at t_0_ = 0 s (to assess baseline concentration of H_2_O_2_) and t_1_ = 60 s (to assess the final concentration of H_2_O_2_) with 100 μL of the Co(NO_3_)_2_ working solution (5 mL of Co(NO_3_)_2_ x 6 H_2_O (0.2 g dissolved in 10 mL ddH_2_O) mixed with 5 mL of (NaPO_3_)_6_ (0.1 g dissolved in 10 mL ddH_2_O) added to 90 mL of NaHCO_3_ (9 g dissolved in 100 mL ddH_2_O)). The H_2_O_2_ concentration in each well was determined indirectly by measuring the rate of oxidation of cobalt (II) to cobalt (III) in the presence of bicarbonate ions based on the absorbance of the carbonato-cobaltate (III) complex ([Co(CO_3_)_3_]Co) at 450 nm using the Infinite F200 PRO multimodal microplate reader (Tecan, Switzerland). The concentration of H_2_O_2_ was determined from the model based on the standard curve obtained from serial dilutions of H_2_O_2_ in 1 × PBS.

### 2.9. Nicotinamide Adenine Dinucleotide Phosphates

Reduced and oxidized fractions of nicotinamide adenine dinucleotide phosphates (NADPH/NADP) were measured using the NADP/NADPH Quantitation kit (Sigma-Aldrich, USA) adhering to the manufacturer’s instructions. To detect NADPH, NADP was decomposed by heating sample aliquots at 60 °C for 30 min.

### 2.10. Data Analysis

Data were analyzed in R (4.1.0) following the principles of analyzing and reporting data from animal experiments [37] and general guidelines for the communication of scientific evidence [38,39]. No blinding was used throughout the experiment. Animals were assigned to groups by stratified randomization to control for the potential effect of body mass and housing in a way that home cage allocation was equally represented in each group and that body mass was balanced between treatments. Redox homeostasis-related variables were analyzed using linear regression with the variable of interest used as the dependent variable and group/treatment allocation and protein concentration (loading control) used as independent variables. Where appropriate additional parameters were included in the model as reported (e.g., baseline H_2_O_2_ concentration in models pertaining to CAT activity, NADP in models pertaining to NADPH, baseline THB absorbance in SOD activity models). Model assumptions were checked using visual inspection of residual and fitted value plots and log-transformation was used where appropriate. Model outputs were reported as point estimates with 95% confidence intervals accompanied by group differences of estimated marginal means or group ratios (for log-transformed dependent variables) with respective 95% confidence intervals. Raw *p*-values were reported alongside estimates. Although it was not possible to assess the sample size *a priori* due to the primary outcome (redox system) being a “composite” of OS-related parameters measured with independent techniques, individual biochemical measurements were adapted to meet optimal sensitivity given available specimens (e.g., larger volumes of samples, longer extraction times, more flashes per well) and uncertainty parameters were communicated alongside estimates wherever possible. Furthermore, although the trends over time were in our focus (and it is not unusual to report the trends in time with independent groups of animals due to a high level of similarity of genetic and environmental factors), we decided to report the results based on group/time-point differences as (i) it was permitted by the design, (ii) we had no sensible time-course model that would assume reasonable changes of variables of interest as a function of the exposure time, (iii) we considered that this “conservative” approach would be more transparent and informative (communicating large uncertainties arising from a small number of animals per group and independence of measurements in time), but not at the expense of appreciation of temporal trends of both monotonic and non-monotonic relationships given appropriate reporting and visualization. For the aforementioned reasons, “statistical significance” between groups/time-points was still reported (α = 5%), although its value was limited in this context. Principal component analysis of centered and scaled model-derived estimates was performed for multivariate exploration. The results of multivariate analyses were represented as the first principal component-ordered Spearman’s rank correlations, biplots of individuals, and vectors. Scree plots and contributions of individual variables are available in the Appendix A.

## 3. Results

### 3.1. The Effect of Acute Oral Galactose Administration on Lipid Peroxidation and Reductive Capacity in Duodenum and Ileum

The administration of galactose solution (200 mg/kg) via the orogastric gavage induced a time-dependent suppression of lipid peroxidation in the duodenum with the lowest concentration of TBARS observed 1 h after the treatment (0–60 CI: 0.38–20.86; t_17_ = 2.19; p_raw_ = 0.043) (Figure 1B). In the ileum, a similar pattern was observed with the lowest values measured 1 h upon galactose administration (0–60 CI: 0.47–30.79; t_17_ = 2.18; p_raw_ = 0.044) (Figure 1D). In order to better understand the context of the effect of galactose on the concentration of lipid peroxidation end products, the overall tissue redox balance was assessed by measuring reductive capacity using NRP [28]. NRP was largely unchanged in both the duodenum and ileum, indicating that the observed reduction of lipid peroxidation did not affect the overall redox balance.

### 3.2. The Effect of Acute Oral Galactose Administration on Low-Molecular-Weight Thiols, Protein Sulfhydryl Content, and Nicotinamide Adenine Dinucleotide Phosphates

The biochemical background accompanying suppressed lipid peroxidation was further explored by quantification of LMWT, determination of reactive protein SH content, and assessment of the cellular pool of NADPH and NADP. Duodenal LMWT demonstrated a tendency of decrement over time that did not meet the criteria of significance given the group size and variance in any group in comparison to t = 0 (Figure 2A,C). Both LMWT and protein SH were reduced by the treatment in the ileum (Figure 2D–F), suggesting a more pronounced consumption of mediators of the nucleophilic arm of redox homeostasis.

To better understand the background of the observed consumption of mediators of the nucleophilic tone, the concentration of NADPH, the most important reducing equivalent for the redox reactions that enable regeneration of glutathione (GSH) was assessed in the same samples. The effect of galactose on NADPH in both the duodenum and ileum (Figure 3A,D,G,H) followed a similar pattern as observed for LMWT—a tendency of decrement over time that did not meet the criteria of significance in any group in contrast to baseline given the group size and variance in the duodenum (Figure 3A), and a similar but more pronounced pattern (treatment-induced decrement) in the ileum (Figure 3D). The effect was maintained after accounting for tissue concentration of NADP (Figure 3C,F–H). Given that NADPH is a substrate for glutathione reductase (GR; EC 1.8.1.7) that catalyzes the reduction of GSSG to GSH [40], and that GSH is the predominant LMWT in animal cells [41], we hypothesized that the observed treatment-induced reduction of both LMWT and NADPH was indicative of the same biological phenomenon that triggered consumption of cellular nucleophiles. Interestingly, although a positive association between the concentration of LMWT and NADPH was observed in the ileum (Figure 3J), this was not the case in the duodenum (Figure 3I), either due to the fact that the observed effects were too small and “masked” by large variance or due to galactose-induced NADPH-GSH uncoupling (e.g., increased consumption of NADPH in pathways that do not generate GSH).

### 3.3. Hydrogen Peroxide Concentration, Hydrogen Peroxide Dissociation Rate, and Superoxide Dismutase Activity

Cellular LMWTs are important substrates for glutathione peroxidases (GPx; EC 1.11.1.9), an enzyme family that protects the cells from oxidative damage by catalyzing the reduction of H_2_O_2_ to H_2_O and reduction of lipid hydroperoxides to alcohols. H_2_O_2_ dissociation rate was increased in the duodenum 60 min after galactose administration (0/60 CI: 0.54–1; t_16_= −2.10; p_raw_ = 0.051|30/60 CI: 0.49–0.95; t_16_= −2.46; p_raw_ = 0.025) (Figure 4A), while peroxidase potential remained unchanged in the ileum (Figure 4C). The baseline concentration of H_2_O_2_ used as a proxy measure for cellular H_2_O_2_ indicated no pronounced galactose-induced alterations in the duodenum (Figure 4B) and a tendency of decrement that did not meet the criteria of significance in any group in comparison to baseline given the variance in the ileum (Figure 4D). The overall SOD activity was increased by galactose treatment in the duodenum (Figure 4E), while a trend indicating suppressed activity was observed for Mn-SOD (Figure 4F). In the ileum, a trend towards increased overall SOD activity was observed in the 60 min and 120 min time-point (Figure 4G), while a trend towards the increased activity of Mn-SOD was observed only in the 120 min time-point (Figure 4H).

### 3.4. Overall Redox-Related Changes in the Duodenum and Ileum following Acute Oral Administration of Galactose

Overall redox-related changes were analyzed by Spearman’s rank correlation of model-derived estimates (Figure 5). Plasma TBARS measured previously [3] was included in the analysis to better understand whether the observed changes reflected systemic or local redox alterations. Time was included in the analysis as a continuous variable to approximate acute galactose-induced changes. It has to be emphasized that such a simplified model was appropriate for monotonic (e.g., duodenal LMWT (Figure 2A)) but not for non-monotonic relationships (e.g., duodenal NADP (Figure 3B)) and should be interpreted with caution. Given the biological context (relatively short time-course data of the biological effects of galactose), the observed monotonic relationships (emphasized by this approach) are more likely to represent important alterations of the biological system. The results of rank correlations were ordered by the first principal component to facilitate the perception of clustered variables. In the duodenum, the strongest clustering effect was observed for the concentration of tissue NADPH, NADP-corrected NADPH, NADP, and MDA (TBARS) (Figure 5A). The greatest association for plasma MDA (ρ = 0.51) was obtained for NADP-corrected NADPH, while a weaker association was observed between plasma MDA and duodenal MDA (ρ = 0.40) (Figure 5A). The strongest association for tissue redox capacity approximated by NRP was observed for NADP (ρ = 0.81) and THB (ρ = −0.66) indicating greater tissue reductive capacity was associated with greater activity of SOD (Figure 5A). A consistent negative association was observed between time since the exposure and mediators of the nucleophilic arm of redox homeostasis—NADP−0corrected NADPH (ρ = −0.51), NADPH (ρ = −0.47), GSH (LMWT) (ρ = −0.59), and SH (ρ = −0.40). At the same time, galactose induced an increase in SOD (THB ρ = −0.68) and CAT activity (ρ = 0.58) and resulted in decreased lipid peroxidation (MDA ρ = −0.52), indicating the observed wasting of nucleophilic substrates was more likely associated with activation of redox defense mechanisms than by redox dyshomeostasis driving the cells towards OS. In the ileum, a similar clustering effect of the mediators of the nucleophilic arm was observed, however, unlike in the duodenum, tissue LMWT (GSH), and SH demonstrated similar loadings as NADP and NADPH (Figure 5B), possibly indicating a greater perturbance of redox homeostasis since wasting of the nucleophilic substrates was more pronounced in the ileum (Figure 2 and Figure 3). Similarly as was observed in the duodenum, there was a negative association of time since the exposure and mediators of the nucleophilic arm of redox homeostasis—SH (ρ = −0.67), GSH (LMWT) (ρ = −0.58), NADP (ρ = −0.71), NADPH (ρ = −0.71), NADP−0corrected NADPH (ρ = −0.62) in the ileum. Furthermore, similarly as observed in the duodenum, the time since the exposure was positively associated in the ileum with the activity of SOD (although a stronger association was observed for mitochondrial Mn−0SOD (THB ρ = −0.30; THB KCN ρ = −0.42)) and negatively with tissue MDA (ρ = −0.76) (Figure 5B). Inversely to what was observed in the duodenum, there was a positive association of catalase activity with the concentration of MDA (ρ = 0.59), and a negative association with time (ρ = −0.56) (Figure 5B) found in the ileum. PCA was used to provide a complementary multivariate exploration of intestinal redox−0related parameters following galactose administration (Figure 5C–F). In the duodenum, galactose treatment was aligned with the second principal component receiving the largest vector projections from CAT (15.7%), NADPH (14.8%), and NADP-corrected NADPH (14.7%), and MDA (12.3%) (Figure 5C,E). In the context of time since the exposure, the largest offset on the second component was observed between the 30 and 60 min time-points (Figure 5C). A change between both the 0 and 30, and 60 and 120 time-points was aligned with the first component with the positive direction of the vector for the early-phase changes (0 to 30) and the negative direction vector for the late-stage changes (60 to 120) (Figure 5C). NADP was the variable contributing the most to the first component with 19.6%, while baseline H_2_O_2_ and NRP contributed 15.4 and 12.6%, respectively, (Figure 5C,E). In the ileum, both galactose treatment and galactose-induced changes over time were exclusively captured by the first component (53.6% of the variance in comparison with 14.9% explained by the second component) (Figure 5D,F). The largest contributors to the first dimension in the ileum were: protein SH (12.9%), NADPH (12.7%), GSH (LMWT) (12.7%)—all biological components of the system responsible for the maintenance of the nucleophilic tone. Plasma MDA insignificantly contributed to the first and second components in the duodenum (3.3% and 1.2%, respectively) and ileum (0.5% and 0%), indicating that the observed changes were reflecting local and not systemic redox alterations.

Rank correlations were then analyzed across tissues to explore the associations of redox-related changes in the proximal small intestine (duodenum) in the context of those observed in the distal segment (ileum). Rank correlation analysis indicated a consistent pattern of a negative association between time after galactose administration and LMWT(GSH) (ρ = −0.57), lipid peroxidation (ρ = −0.41), and NADP (ρ = −0.37) (Figure 6A). Inversely, time after galactose administration was positively associated with the overall activity of SOD (THB ρ = −0.37) across tissues (Figure 6A). The overall reductive capacity of tissue (NRP) was positively associated with the activity SOD (ρ = 0.54); however, there was a negative association with protein SH (ρ = −0.45) where NRP sensitivity towards reactive mediators of redox homeostasis may be one possible explanation. There was a negative association between tissue MDA and the overall activity of SOD (THB ρ = 0.50), possibly reflecting the role of superoxide radicals in the formation of lipid peroxyl radicals—a critical step of the propagation phase of lipid peroxidation [42]. Furthermore, there was a positive association of tissue MDA and mediators of the nucleophilic arm of redox homeostasis—NADP−0corrected NADPH (ρ = 0.77), and NADPH (ρ = 0.74), possibly mirroring the overall pattern of galactose−0induced activation of nucleophilic mediator-dependent mechanisms that resulted in the suppression of lipid peroxidation (Figure 6A). PCA was used to provide a complementary multivariate exploration of intestinal redox-related parameters following galactose administration (Figure 6B,C). Here, the effects of the anatomical segment (Figure 6B, upper panel), galactose treatment (Figure 6B, middle panel), and time since the exposure (Figure 6B, lower panel) were all clearly delineated in respect to the first two components explaining 45 and 17.2% of the variance, respectively. NADPH, NADP-corrected NADPH, and baseline H_2_O_2_ were the largest contributors to the first principal component with 15.3%, 14.6%, 14%, respectively. The main contributor to the second component was LMWT (GSH) (28.3%), followed by SH (16.3%), Mn-SOD (THB KCN) (15.4%), and NADP (12.6%) (Figure 6B,C). An inverse NADP vector was the most aligned with the temporal vector indicating NADP reduction as a valuable predictor of assignment of individuals to groups/time-points (Figure 6C).

## 4. Discussion

Biological context should be taken into account and clearly delineated in order to understand the observed findings. First, it should be acknowledged that the maximal local exposure to galactose in the duodenum and the ileum correspond to different time-points. The solution administered via the orogastric catheter was expected to reach the duodenum before the first time-point used in the study (t = 30 min), and the residual solution was expected to reach the distal ileum before the final time-point (t = 120 min) [43], so the effect of maximal local exposure to galactose solution is expected in the 30 min time-point in the duodenum, and in the 120 min time-point in the ileum. Second, it has to be emphasized that the effective exposure was likely different in the duodenum and the ileum as galactose was absorbed along its way through the small intestine. Furthermore, as the tissue exposure took place inside a complex biological system (in the rat gastrointestinal tract), it should be taken into account that systemic changes (e.g., secondary to the effect of the solution in the duodenum) affected both the duodenum and the ileum at the same time. Although simpler solutions (e.g., ex vivo experiments) might have provided a “cleaner” insight into the effects of galactose on redox homeostasis in individual segments of the rat small intestine not complicated by the interdependence of organismic physiological systems, the *in vivo* approach was used to obtain data relevant in the context of its potential application—understanding the effects of orally administered galactose. Finally, the administered galactose solution (1 mL of 0.55 M galactose) most likely exceeded the local tissue metabolic and active transport capacity for galactose. There is no data on the exact metabolic and absorptive capacity for d-galactose in individual segments of the rat small intestine and the assumption of saturated metabolic and absorptive capacity remains only hypothetical. Nevertheless, given the very large concentrations (0.55 M) administered and the fact that active transport of glucose (mediated by the same mechanism) becomes saturated at “high luminal concentrations” greater than 30 mM [44], we consider oversaturation highly plausible.

The reported results suggest that acute oral administration of d-galactose (200 mg/kg) can suppress intestinal lipid peroxidation with the lowest concentration of oxidized products (TBARS) observed one hour after orogastric gavage in both the duodenum and ileum (Figure 1B). Galactose-induced decrement of lipid peroxidation was not accompanied by pronounced alterations of overall reductive capacity suggesting modest alterations of the overall redox balance (Figure 1C,E). In the process of lipid peroxidation, after primary peroxidation end products are produced, cells utilize two-electron reduction of hydroperoxides catalyzed by GPx to prevent one-electron reduction decomposition that supports the propagation of lipid peroxidation [42]. Consequently, GSH (the main LMWT) used by GPx for reduction of hydroperoxides, protein SH (that form intra-protein disulfides or undergo S-thiolation in response to redox perturbations [45]), and NADPH (that takes part in the restoration of GSH stores acting as a substrate for GR) were examined to assess whether the activation of the biochemical system involved in peroxidative damage inhibition might have been the mediator of the observed galactose-induced decrement of lipid peroxidation end products. Both LMWT and NADPH concentrations (total and corrected for the concentration of NADP) were reduced by the treatment indicating galactose might have activated peroxidative damage defense pathways. Furthermore, a similar pattern was observed for protein SH in the ileum suggesting that the protein thiol redox state might have been affected by the treatment (Figure 2E). However, in the duodenum, a trend of reduced SH was only present 2 h after the treatment (Figure 2B), indicating a less pronounced effect on the duodenal redox system. The explanation for the observed difference can only be speculated; however, the association between the concentration of LMWT and NADPH in the ileum (Figure 3J) and the lack thereof in the duodenum (Figure 3I) suggests the difference might be due to biochemical uncoupling of the reactions generating/consuming NADPH and those generating/consuming GSH. A similar positive association was observed for protein sulfhydryl groups in the ileum where NADPH was predictive of protein SH (R^2^ = 0.34), while the association was absent in the duodenum (R^2^ = 0.01). Another explanation might be that the galactose-induced challenge to redox homeostasis was simply more pronounced in the ileum and alterations in the thiol redox state were too subtle to be detected with DTNB that reacts with thiols independently of their ionization state [45]. The latter is possible as all redox-related changes were generally more pronounced in the ileum.

Following the general impression that d-galactose might have decreased TBARS by activating redox systems involved in peroxidative damage protection that in turn consumed LMWT and consequently also NADPH, we examined whether the response to d-galactose was associated with alterations of the H_2_O_2_ dissociation rate as H_2_O_2_ is another major redox metabolite operative in redox signaling and regulation under the control of GPx [46]. Furthermore, H_2_O_2_ is an important metabolic signaling molecule acting as a second messenger of insulin signaling [46], playing a key role in the famous “peroxide dilemma” [47], and is tightly associated with nutrient sensing and mitochondrial function [46]. Interestingly, although an inverse relationship was found between the concentration of TBARS and H_2_O_2_ dissociation capacity in the duodenum (Figure 1B, 4A and 5A), such a trend was not observed in the ileum. As one of the major functions of GPx is to catalyze the reduction of H_2_O_2_ to H_2_O, the increased dissociation capacity in the duodenum might provide an explanation for the observed discrepancies between the relationship of LMWT/SH and NADPH in the duodenum and in the ileum assuming additional LMWT were consumed for GPx-mediated control of H_2_O_2_ homeostasis in the ileum to compensate for the absence of upregulation of H_2_O_2_ dissociation capacity. Inversely, it is possible that increased wasting of substrates required for the maintenance of the nucleophilic tone (LMWT and NADPH) incapacitated the response in the ileum due to a limited pool of substrates for GPx-mediated regulation of H_2_O_2_.

Finally, to better understand the observed increment of H_2_O_2_ dissociation rate, we aimed to assess the activity of SODs as a major source of H_2_O_2_ produced in the SOD-catalyzed dismutation of the superoxide radical anion to O_2_ and H_2_O_2_ [46]. Interestingly, galactose treatment was associated with a trend towards increased total SOD activity both in the duodenum and in the ileum (Figure 4E,G). Considering that superoxide anions have been proposed as the origin of most cellular reactive oxygen and nitrogen species playing a central role in redox homeostasis and OS [48], the detected tendency of galactose to potentiate the activity of SODs may provide an explanation for the observed decrease of lipid peroxidation. As potentiation of SOD activity by delivering SODs or catalytic SOD mimetics seems to be a promising pharmacological goal for many OS-related diseases [49], a possibility that d-galactose (given in the right dose) may potentiate SOD activity should be further explored using appropriate sample sizes and time-points. Unlike a comparable pattern of the trend towards activation observed for total SOD in both the duodenum and ileum, d-galactose treatment reduced the activity of Mn-SOD in the duodenum but increased it in the ileum (Figure 4F,H). This perplexing discrepancy remains to be understood.

An alternative explanation for the observed galactose-induced reduction of lipid peroxidation is the possibility that excessive local concentration of galactose saturated the metabolic machinery and fueled alternative metabolic pathways—mainly the production of galactitol by AR, and D-galactonate produced by galactose dehydrogenase (EC 1.1.1.48). Hypothetically, activation of AR could explain the observed consumption of NADPH, and it is theoretically possible that potentiation of enzymatic activity might have also resulted in a reduced concentration of TBARS as MDA oxidation by mitochondrial AR has been proposed as a probable metabolic route [42]. Although the competition of substrates (galactose and MDA) makes this explanation unlikely, if it proves to be correct it may shine a new light on the possible repercussions of the activation of AR especially in the context of galactitol toxicity that has been questioned since the concentration produced in the lens of rats on a very high galactose diet may not be high enough to cause osmotic stress [50].

Regardless of the mechanisms responsible for the observed effects of d-galactose on redox homeostasis of the duodenum and ileum the presented results provide foundations for further exploration of potentially protective action of galactose in the gastrointestinal tract as the affected redox subsystems (SODs, GPx, and lipid peroxidation) play a substantial role in the maintenance of gastrointestinal health.

The intestine is the primary organ for food absorption, and metabolism, and an essential immunological and physical barrier inevitably exposed to foreign substances and microorganisms with substantial biochemical potential for producing electrophiles, which was recognized as a “free radical time bomb” a long time ago [51]. Redox homeostasis plays a critical role in the maintenance of the structure and function of the gastrointestinal barrier [25]. Indomethacin-induced gastrointestinal mucosal injury can be prevented by pretreatment with SOD [52], decreased intestinal SOD has been associated with the formation of gastric ulcers, and its increased activity was observed in the process of ulcer healing [25,53]. The activity of GPx that seems to be potentiated by galactose is critical for the maintenance of the gastrointestinal barrier by protecting the gut against the absorption of harmful dietary hydroperoxides and the effects of microorganism-stimulated gastrointestinal reactive oxygen species [25,54,55]. Furthermore, GPx knockout mice for the widely expressed GPx1 and its epithelium-specific variant GPx2 develop intestinal inflammation and colitis [56]. Finally, uncontrolled generation of intestinal electrophiles due to inflammation, dysbiosis, or ingestion of alcohol, herbicides, dietary iron, or certain medications (e.g., non-steroidal anti-inflammatory drugs) [57] initiates lipid peroxidation, resulting in the production of primary and secondary peroxidation products [42]. The concept that disbalance between electrophilic and nucleophilic tone encourages initiation and propagation and delays the termination step of lipid peroxidation is central to most theories of OS-induced pathophysiological states (at least those predominantly defined by the excess of electrophiles [58,59]) and accumulation of lipid peroxidation end products is evident in many pathological states including those affecting the gastrointestinal tract (e.g., reflux esophagitis, gastritis, gastric ulcers, and *H. pylori* infection, and inflammatory bowel disease [25]).

Considering that gastrointestinal dyshomeostasis and inflammation have been recognized as important factors that may be involved in the etiopathogenesis of neuroinflammation and degenerative changes in the central nervous system [60], the effects of galactose on intestinal redox homeostasis should be further explored in a design using long-term administration of d-galactose dissolved in drinking water that has previously shown neuroprotective effects [3,15,16,17]. The latter may be especially important as chronic oral administration of relatively low doses of d-galactose (100 mg/kg) were reported to induce cognitive impairment and OS when given by oral gavage in a bolus dose once per day [19]. In contrast, even very large doses of d-galactose were not able to induce inflammation in rats when the administration was spread throughout the day [61]. In an attempt to induce inflammation and adverse metabolic effects to test the anti-inflammatory potential of fructooligosaccharides added to the dietary regimen, Mhd Omar et al. fed the rats with a high galactose diet (50 g/100 g) for 12 weeks [61]. As rats need approximately 15 g of standardized food pellets per day [62], the rats fed a high galactose diet ingested ~7.5 g of galactose per day—an approximately 150 times greater dose in comparison to the one that caused OS and cognitive impairment when administered in the form of a repeated orogastric bolus dose [19]. Even with such a high concentration of galactose, there was no change in plasma zonulin, interleukin-1β, interleukin-6, tumor necrosis factor-α, and C-reactive protein in comparison to that observed in rats fed with the isocaloric control diet (61.5 g/100 g starch) [61]. Furthermore, 12 weeks of high galactose diet reduced body mass gain and galactose-treated rats had ~55% reduced concentration of plasma endotoxin indicating improvement of the gastrointestinal barrier function [61]. Why oral galactose can improve intestinal barrier function remains to be explored, but given its role in glycan synthesis, one possibility is that it can support the production of mucins and integrity of the mucus barrier that acts as one of the first lines of protection of the gastrointestinal tract [63,64], which has been recognized for its potential importance in neurological disorders [65]. Another possibility, based on the findings reported here, is that galactose is able to support intestinal redox homeostasis and maintain integrity of the barrier despite constant exposure of intestinal epithelial cells to harmful dietary hydroperoxides and the effects of microorganism-stimulated gastrointestinal OS.

## 5. Conclusions

The effects of d-galactose on redox systems are far from trivial and simple explanations suggesting monotone pro-oxidative effects regardless of the time/dose-response, route of administration, region-dependent metabolic capacities, and the presence/absence of underlying pathophysiological processes probably provide poor estimates of its true biological effects. Based on the findings from the acute time-response experiment presented here, it seems that the redox system of the small intestine can acutely tolerate even high luminal concentration of galactose (0.55 M), and that galactose may increase the activity of SODs, and suppress lipid peroxidation and/or potentiate the removal of harmful electrophilic aldehydes (possibly via GPx) consuming the pool of nucleophilic substrates. The redox modulating ability of oral d-galactose should be further explored, especially in the context of maintenance of the integrity of the intestinal barrier and its possible implications for beneficial cognitive effects of long-term administration of galactose dissolved in drinking water.

## 6. Limitations

This study was exploratory in its design and the obtained results should be interpreted in this context—as hypotheses-generating rather than hypothesis-testing data. As only six animals were used per group, it is rational to conclude the optimal power to detect all redox-related parameters measured was not achieved, so we did not focus on *p*-values, but provided comparisons between groups/time-points as plots of the effects. Although it is not unusual to model parameters of interest using exposure time as the independent variable in experimental designs similar to the one used in this study, we decided to use a conservative approach and model time-points as groups to (i) emphasize uncertainties introduced by small sample size and independence of measurements in regards to time-points (although animals were randomized to treatment groups/time-points data-points still represent individual animals exclusively in a single time-point), and (ii) address the problem of a non-existent sensible time-course model that would assume reasonable changes of variables of interest as a function of the exposure time (ideally based on previous experimental data from the literature that unfortunately does not exist). Model outputs were reported in a way that still enables the evaluation of temporal trends, and rank-correlations were reported for model estimates to further emphasize monotonic trends (that may be biologically most valuable considering the relatively short time-course (120 min) used in the study). An alternative experimental design that would have enabled collection of repeated measures of the effects of acute oral galactose administration on redox homeostasis was also considered (by placing a microdialysis probe in the area of interest); however, this option was abandoned due to the observed effects of microdialysis procedure on the parameters involved in the regulation of redox homeostasis [3].

Finally, the results presented here reflect the effects of a single oral bolus dose of d-galactose and cannot be directly translated to the effects of the chronic exposure to low dose d-galactose achieved by *ad libitum* treatment that has shown neuroprotective effects [15,16,17]. Nevertheless, the presented results clearly show that even intraluminal concentrations much larger than those achieved by the dissolution of d-galactose in drinking water are not able to saturate the intestinal metabolic and redox capacity and induce OS. Considering that rats consume approximately 10 mL of water/100 g body weight/day, animals treated with 200 mg/kg d-galactose dissolved in drinking water available *ad libitum* are repeatedly exposed to intraluminal concentrations of ~2 mg/mL d-galactose (~40 times lower than the concentration tested here). Given a very large safety margin, from a toxicological perspective, we consider that it is not very likely that harmful effects on intestinal redox homeostasis will be observed even after chronic exposure. On the other hand, the existence of beneficial effects on redox homeostasis might also be dose-dependent, and it remains to be tested whether chronic exposure to low dose d-galactose can alleviate intestinal OS as can be hypothesized from the presented results.

## Figures and Tables

**Figure 1 antioxidants-11-00037-f001:**
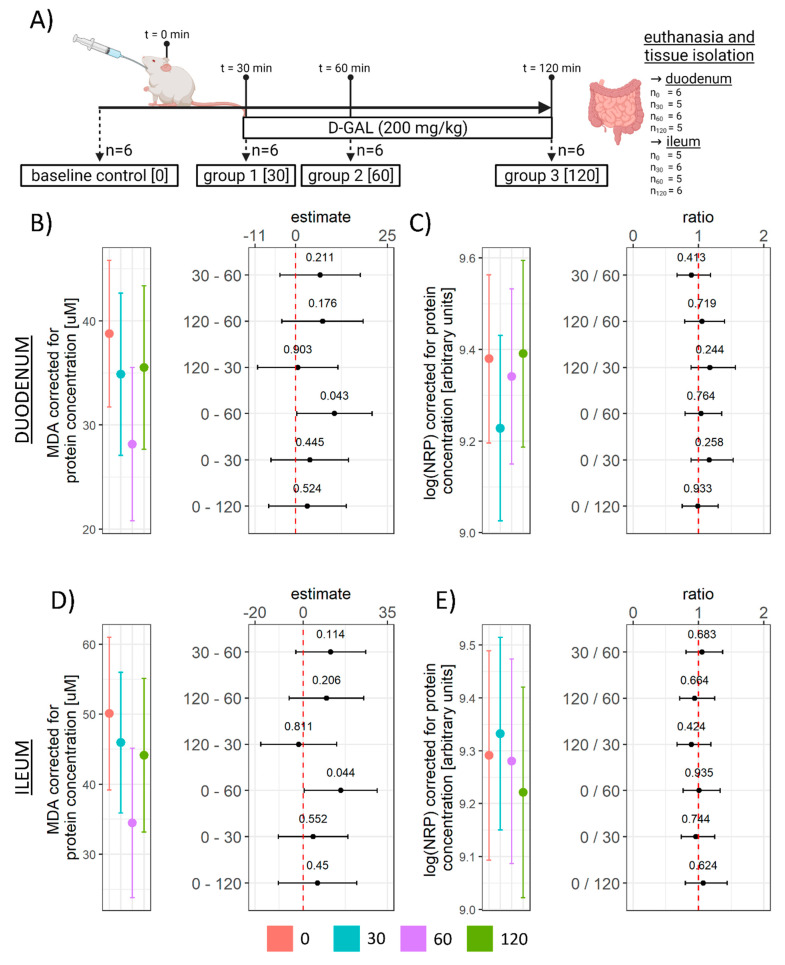
Experimental design, the concentration of TBARS, and reductive capacity in the rat duodenum and ileum following acute orogastric gavage of 200 mg/kg d-galactose. (**A**) Experimental design. One group of animals was sacrificed before the treatment and served as the baseline control (0; *n* = 6), 3 groups of animals received 1 mL of galactose solution (200 mg/kg dissolved in water) by orogastric gavage and were sacrificed after 30 (30; *n* = 6), 60 (60; *n* = 6), and 120 (120; *n* = 6) minutes. (**B**) Duodenum TBARS model output (group point estimates with 95% confidence intervals) (**left**) and the differences of estimated marginal means for groups with corresponding 95% confidence intervals (**right**). (**C**) Duodenum log(NRP) model output (group point estimates with 95% confidence intervals) (**left**) and ratios of estimated marginal means for groups with corresponding 95% confidence intervals (**right**). (**D**) Ileum TBARS model output (group point estimates with 95% confidence intervals) (**left**) and the differences of estimated marginal means for groups with corresponding 95% confidence intervals (**right**). (**E**) Ileum log(NRP) model output (group point estimates with 95% confidence intervals) (**left**) and ratios of estimated marginal means for groups with corresponding 95% confidence intervals (**right**). Raw *p*-values (black) are reported alongside point estimates. TBARS—Thiobarbituric acid reactive substances; NRP—Nitrocellulose redox permanganometry. 0—Baseline control; 30—group sacrificed 30 min after galactose treatment; 60—group sacrificed 60 min after galactose treatment; 120—group sacrificed 120 min after galactose treatment.

**Figure 2 antioxidants-11-00037-f002:**
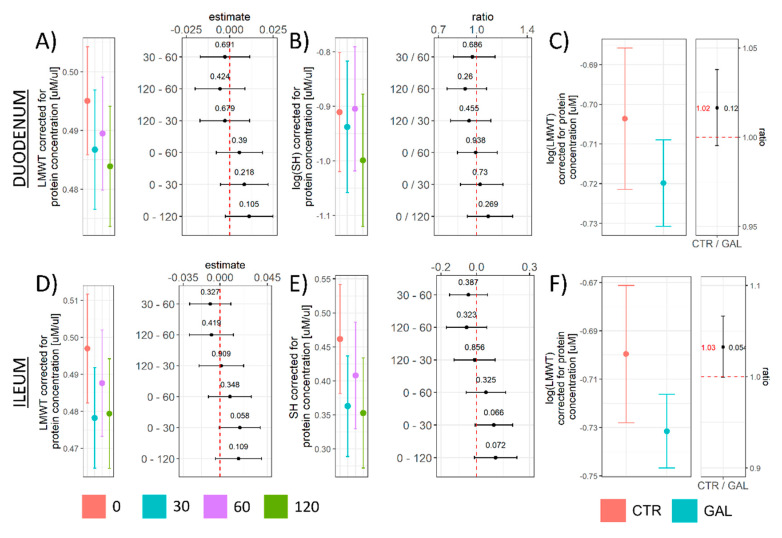
The concentration of LMWT and tissue protein SH content in the rat duodenum and ileum following acute orogastric gavage of 200 mg/kg d-galactose. (**A**) Duodenum LMWT model output (group point estimates with 95% confidence intervals) (**left**) and the differences of estimated marginal means for groups with corresponding 95% confidence intervals (**right**). (**B**) Duodenum protein SH model output (group point estimates with 95% confidence intervals) (**left**) and ratios of estimated marginal means for groups with corresponding 95% confidence intervals (**right**). (**C**) Duodenum log(LMWT) pooled treatment effect model output (group point estimates with 95% confidence intervals) (**left**) and ratios of estimated marginal means for treatments with corresponding 95% confidence intervals (**right**). (**D**) Ileum LMWT model output (group point estimates with 95% confidence intervals) (**left**) and the differences of estimated marginal means for groups with corresponding 95% confidence intervals (**right**). (**E**) Ileum protein SH model output (group point estimates with 95% confidence intervals) (**left**) and the differences of estimated marginal means for groups with corresponding 95% confidence intervals (**right**). (**F**) Ileum log(LMWT) pooled treatment effect model output (group point estimates with 95% confidence intervals) (**left**) and ratios of estimated marginal means for treatments with corresponding 95% confidence intervals (**right**). LMWT—low-molecular-weight thiols; SH—protein sulfhydryl content. Raw *p*-values (black) and raw values (red) are reported alongside point estimates. 0—Baseline control; 30—group sacrificed 30 min after galactose treatment; 60—group sacrificed 60 min after galactose treatment; 120—group sacrificed 120 min after galactose treatment. CTR—Control animals; GAL—animals treated with galactose.

**Figure 3 antioxidants-11-00037-f003:**
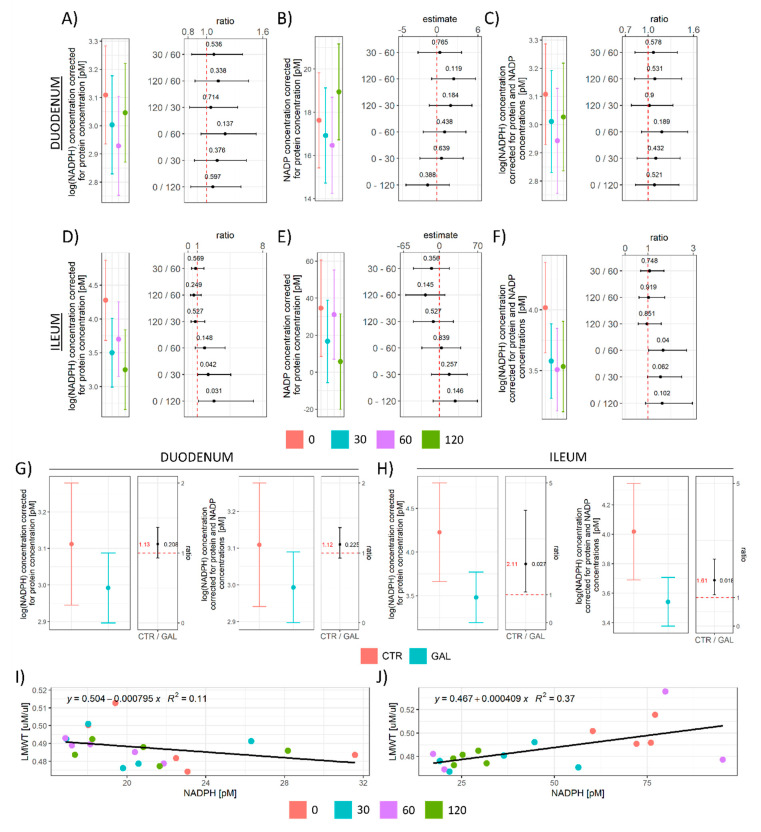
The concentration of reduced and oxidized nicotinamide adenine dinucleotide phosphates (NADPH and NADP) in the rat duodenum and ileum following acute orogastric gavage of 200 mg/kg d-galactose. (**A**) Duodenum NADPH model output (group point estimates with 95% confidence intervals) (**left**) and ratios of estimated marginal means for groups with corresponding 95% confidence intervals (**right**). (**B**) Duodenum NADP model output (group point estimates with 95% confidence intervals) (**left**) and differences of estimated marginal means for groups with corresponding 95% confidence intervals (**right**). (**C**) Duodenum NADPH corrected for the cellular pool of NADP. Model output (group point estimates with 95% confidence intervals) (**left**) and ratios of estimated marginal means for groups with corresponding 95% confidence intervals (**right**). (**D**) Ileum NADPH model output (group point estimates with 95% confidence intervals) (**left**) and ratios of estimated marginal means for groups with corresponding 95% confidence intervals (**right**). (**E**) Ileum NADP model output (group point estimates with 95% confidence intervals) (**left**) and differences of estimated marginal means for groups with corresponding 95% confidence intervals (**right**). (**F**) Ileum NADPH corrected for the cellular pool of NADP. Model output (group point estimates with 95% confidence intervals) (**left**) and ratios of estimated marginal means for groups with corresponding 95% confidence intervals (**right**). (**G**) Duodenum log(NADPH) pooled treatment effect model output (group point estimates with 95% confidence intervals) with ratios of estimated marginal means for treatments with corresponding 95% confidence intervals (**left**). Duodenum log(NADPH) pooled treatment effect corrected for the cellular pool of NADP. Model output (group point estimates with 95% confidence intervals) with ratios of estimated marginal means for treatments with corresponding 95% confidence intervals (**right**). (**H**) Ileum log(NADPH) pooled treatment effect model output (group point estimates with 95% confidence intervals) with ratios of estimated marginal means for treatments with corresponding 95% confidence intervals (**left**). Ileum log(NADPH) pooled treatment effect corrected for the cellular pool of NADP. Model output (group point estimates with 95% confidence intervals) with ratios of estimated marginal means for treatments with corresponding 95% confidence intervals (**right**). (**I**) Scatterplot of LMWT and NADPH in the duodenum. (**J**) Scatterplot of LMWT and NADPH in the ileum. Raw *p*-values (black) and raw values (red) are reported alongside point estimates. 0—Baseline control; 30—group sacrificed 30 min after galactose treatment; 60—group sacrificed 60 min after galactose treatment; 120—group sacrificed 120 min after galactose treatment. CTR—control animals; GAL—animals treated with galactose; LMWT—low-molecular-weight thiols.

**Figure 4 antioxidants-11-00037-f004:**
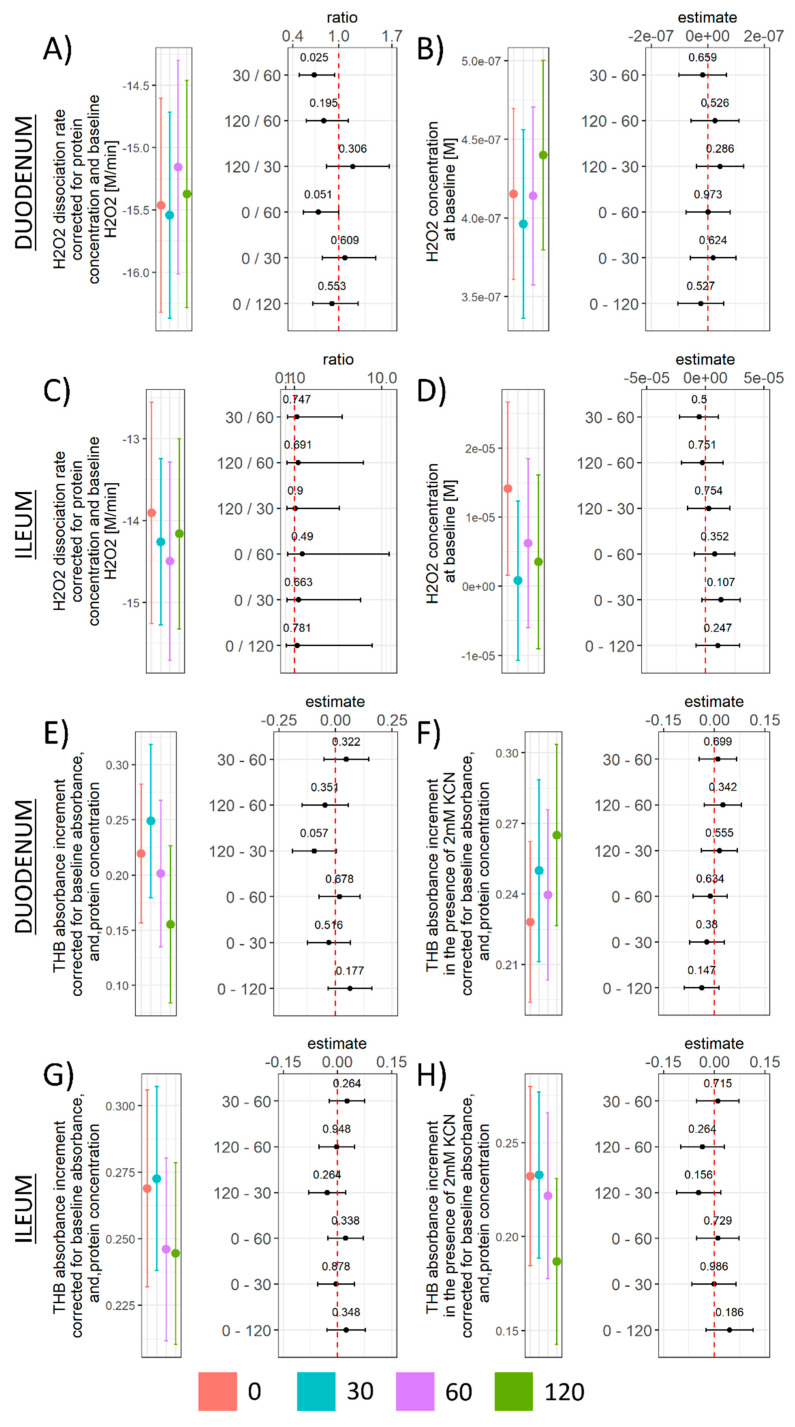
H_2_O_2_ dissociation rate, H_2_O_2_ concentration at baseline, and SOD activity in the rat duodenum and ileum following acute orogastric gavage of 200 mg/kg d-galactose. (**A**) Duodenum H_2_O_2_ dissociation rate model output (group point estimates with 95% confidence intervals) (**left**) and ratios of estimated marginal means for groups with corresponding 95% confidence intervals (**right**). (**B**) Duodenum H_2_O_2_ concentration at baseline model output (group point estimates with 95% confidence intervals) (**left**) and differences of estimated marginal means for groups with corresponding 95% confidence intervals (**right**). (**C**) Ileum H_2_O_2_ dissociation rate model output (group point estimates with 95% confidence intervals) (**left**) and ratios of estimated marginal means for groups with corresponding 95% confidence intervals (**right**). (**D**) Ileum H_2_O_2_ concentration at baseline model output (group point estimates with 95% confidence intervals) (**left**) and differences of estimated marginal means for groups with corresponding 95% confidence intervals (**right**). (**E**) Duodenum THB absorbance increment (inversely proportional to total SOD activity) model output (group point estimates with 95% confidence intervals) (**left**) and differences of estimated marginal means for groups with corresponding 95% confidence intervals (**right**). (**F**) Duodenum THB absorbance increment in the presence of 2 mM KCN (inversely proportional to the activity of Mn-SOD and Fe-SOD due to inhibition of Cu/Zn-SOD) model output (group point estimates with 95% confidence intervals) (**left**) and differences of estimated marginal means for groups with corresponding 95% confidence intervals (**right**). (**G**) Ileum THB absorbance increment (inversely proportional to total SOD activity) model output (group point estimates with 95% confidence intervals) (**left**) and differences of estimated marginal means for groups with corresponding 95% confidence intervals (**right**). (**H**) Ileum THB absorbance increment in the presence of 2 mM KCN (inversely proportional to the activity of Mn-SOD and Fe-SOD due to inhibition of Cu/Zn-SOD) model output (group point estimates with 95% confidence intervals) (**left**) and differences of estimated marginal means for groups with corresponding 95% confidence intervals (**right**). H_2_O_2_—hydrogen peroxide; SOD—superoxide dismutase; THB—1,2,3-trihydroxybenzene; KCN—potassium cyanide; Mn-SOD—manganese superoxide dismutase; Fe-SOD—iron superoxide dismutase; Cu/Zn-SOD—copper/zinc superoxide dismutase. Raw *p*-values black are reported alongside point estimates. 0—Baseline control; 30—group sacrificed 30 min after galactose treatment; 60—group sacrificed 60 min after galactose treatment; 120—group sacrificed 120 min after galactose treatment.

**Figure 5 antioxidants-11-00037-f005:**
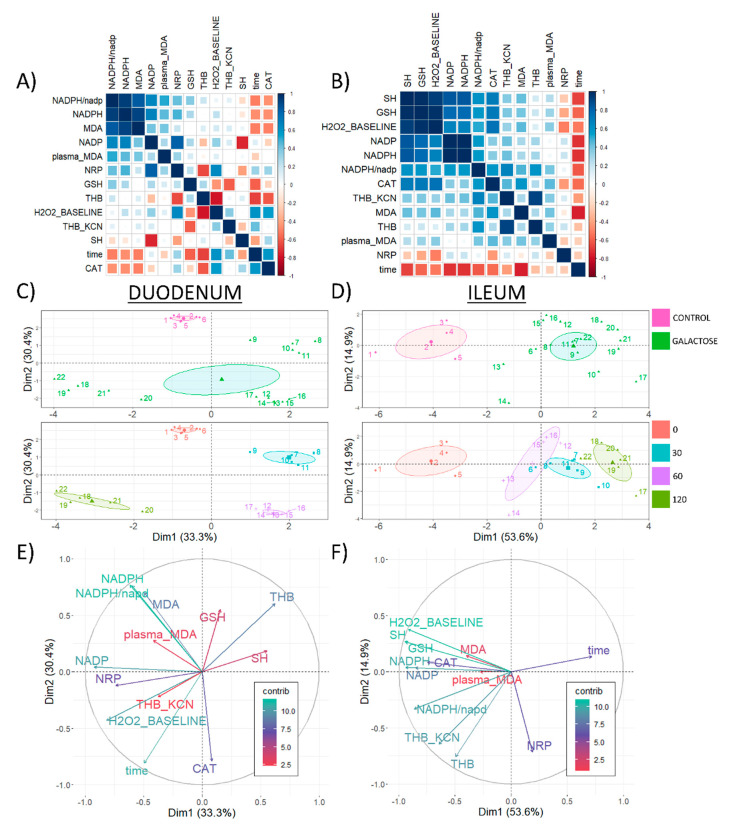
Correlation analysis of redox-related parameters in the rat duodenum and ileum following acute orogastric gavage of 200 mg/kg d-galactose. (**A**) Spearman’s rank correlation of duodenal redox-related parameters ordered by the first principal component. (**B**) Spearman’s rank correlation of ileal redox-related parameters ordered by the first principal component. (**C**) Duodenum principal component analysis biplot indicating coordinates of individual animals in respect to first and second principal components with colors indicating treatment (upper: baseline control vs. galactose treatment) and time since the exposure to orogastric gavage of 200 mg/kg d-galactose solution (lower: baseline control [0] vs. 30 [30] vs. 60 [60] vs. 120 [120] minutes since the exposure). (**D**) Ileum principal component analysis biplot indicating coordinates of individual animals in respect to first and second principal components with colors indicating treatment (upper: baseline control vs. galactose treatment) and time since the exposure to orogastric gavage of 200 mg/kg d-galactose solution (lower: baseline control [0] vs. 30 [30] vs. 60 [60] vs. 120 [120] minutes since the exposure). (**E**) Duodenum biplot graph of variables’ contributions to the 1st principal component (color and vector projection length) and the 2nd principal component (vector projection length). (**F**) Ileum biplot graph of variables’ contributions to the 1st principal component (color and vector projection length) and the 2nd principal component (vector projection length). NRP—nitrocellulose redox permanganometry; time—minutes since the exposure to treatment (continuous variable); THB_KCN—change in 1,2,3-trihydroxybenzene autooxidation rate in the presence of 2 mM potassium cyanide (KCN); GSH—glutathione (the main representative of the low-molecular-weight thiols (LMWT)); MDA—malondialdehyde (the main representative of the thiobarbituric acid reactive substances (TBARS)); NADP—nicotinamide adenine dinucleotide phosphate; THB—change in 1,2,3-trihydroxybenzene autooxidation rate; CAT—catalase activity (hydrogen peroxide dissociation rate); H_2_O_2__BASELINE—estimated baseline levels of tissue hydrogen peroxide; SH—protein reactive sulfhydryl groups; NADPH—reduced nicotinamide adenine dinucleotide phosphate; NADPH/nadp—the concentration of reduced nicotinamide adenine dinucleotide phosphate corrected for concentration of nicotinamide adenine dinucleotide phosphate; contrib—contribution of variables; Dim1—1st dimension; Dim2—2nd dimension.

**Figure 6 antioxidants-11-00037-f006:**
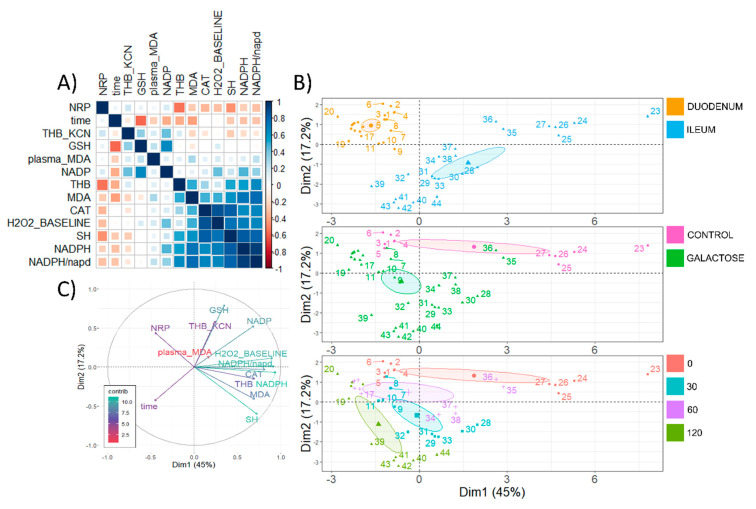
Rank correlation and principal component analysis of rat redox-related parameters in the small intestine following acute orogastric gavage of 200 mg/kg d-galactose. (**A**) Spearman’s rank correlation of redox-related parameters measured in the small intestine (duodenum and ileum) ordered by the first principal component. (**B**) Principal component analysis biplot indicating coordinates of individual animals in respect to 1st and 2nd principal components with colors indicating anatomical segment (upper: duodenum vs. ileum), treatment (center: baseline control vs. galactose treatment), and time since the exposure to orogastric gavage of 200 mg/kg d-galactose solution (lower: baseline control [0] vs. 30 [30] vs. 60 [60] vs. 120 [120] minutes since the exposure). (**C**) A biplot graph of variables’ contributions to the 1st principal component (color and vector projection length) and the 2nd principal component (vector projection length). NRP—nitrocellulose redox permanganometry; time—minutes since the exposure to treatment (continuous variable); THB_KCN—change in 1,2,3-trihydroxybenzene autooxidation rate in the presence of 2 mM potassium cyanide (KCN); GSH—glutathione (the main representative of the low-molecular-weight thiols (LMWT)); MDA—malondialdehyde (the main representative of the thiobarbituric acid reactive substances (TBARS)); NADP—nicotinamide adenine dinucleotide phosphate; THB—change in 1,2,3-trihydroxybenzene auto-oxidation rate; CAT—catalase activity (hydrogen peroxide dissociation rate); H_2_O_2__BASELINE—estimated baseline levels of tissue hydrogen peroxide; SH—protein reactive sulfhydryl groups; NADPH—reduced nicotinamide adenine dinucleotide phosphate; NADPH/nadp—the concentration of reduced nicotinamide adenine dinucleotide phosphate corrected for the concentration of nicotinamide adenine dinucleotide phosphate; contrib—contribution of variables; Dim1—1st dimension; Dim2—2nd dimension.

## Data Availability

Raw data can be obtained from https://github.com/janhomolak (accessed on 23 December 2021).

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
