# Peer review of "The Effect of Acute Oral Galactose Administration on the Redox System of the Rat Small Intestine"

_antioxidants, 2021, doi:10.3390/antiox11010037_

Round 1

Reviewer 1 Report

The subject of the manuscript is very interesting and is within the scope of the Journal. There is a strong need to understand the mechanism of action of galactose towards free radicals generation, and to reveal differences in the biological effects between parenteral and oral administration of that substance. Below are my recommendations to improve the manuscript:

  1. The Introduction part is to long and should be shortened. Moreover, sometimes it is hard to understand in what conditions galactose acts as OS stimulant, and when it provides beneficial effects to cognitive functions. It must be clearly stated (like in the Abstract), that chronic parenteral galactose administration triggers OS, and oral administration provides beneficial effects to the body.
  2. Did the Authors obtain the approval of the appropriate ethic comitee?
  3. The Authors wrote that studies on the role of galactose in animals are bias and their scientific value may be low. But, going through the manuscript I do not see any information how many repetitions of each measurments were performed. How can we be sure that these results are reliable and reproducible?
  4. line 633 - "The intestine is the primary organ for food digestion, absorption, and metabolism". Rather, the first organ involved in digestion is the stomach, not the intestine.
  5. In general, both parts Results and also Discussion, are too extensive, and it is difficult to understand what the research actually shows. Please make it more concise and provide only the most important information resulting from the conducted research.  

Author Response

A point-by-point response to reviewers regarding the manuscript: „ The effect of acute oral galactose administration on the redox system of the rat small intestine“

Reviewer 1

We wish to thank Reviewer 1 for their time and effort in reviewing our manuscript. We strongly believe addressing the comments improved the quality (and reliability) of the data we presented and the quality of the manuscript in general.

Comments and Suggestions for Authors

The subject of the manuscript is very interesting and is within the scope of the Journal. There is a strong need to understand the mechanism of action of galactose towards free radicals generation, and to reveal differences in the biological effects between parenteral and oral administration of that substance. Below are my recommendations to improve the manuscript:

Thank you. We agree that there is a strong need to understand mechanisms mediating the effects of both exogenous (oral and parenteral) and endogenous galactose in the body, especially its effects on redox homeostasis.

The Introduction part is to long and should be shortened. Moreover, sometimes it is hard to understand in what conditions galactose acts as OS stimulant, and when it provides beneficial effects to cognitive functions. It must be clearly stated (like in the Abstract), that chronic parenteral galactose administration triggers OS, and oral administration provides beneficial effects to the body.

Thank you for this suggestion. We completely agree that a shorter and more concise introduction might be easier to follow and might be more suitable in most cases. We originally wrote an extensive introduction because of the paradox that D-galactose is almost exclusively examined in light of its senescence-inducing effects upon continuous (usually) parenteral administration, while its physiological importance, although acknowledged remains largely unexplored. Even more importantly, it seems that relatively scarce studies on the critical physiological (and beneficial) effects of D-galactose fail to acknowledge a large body of evidence suggesting its harmful effects, and vice versa - resulting in two “research bubbles” that will likely not contribute to true understanding of the biological role of this ubiquitous monosaccharide. In our experience, without clearly explaining this “paradoxical research setting” the readers often fail to understand why we would approach the research question the way we did (e.g. why would we want to study the effects of oral administration [physiological route of administration, and the one that has very rarely been reported to cause any harmful effects] on the redox system [that has been proposed as the most likely mediator of the harmful effects]). In the introduction in its present form we:

  1. Briefly introduce D-galactose - the main subject of our research - in its broader context in a few sentences (1st paragraph)

  2. Briefly explain the main findings from galactose-induced aging studies that present a large proportion of scientific literature on D-galactose (2nd paragraph) and comment on why at least some of the claims in this field should be questioned.

  3. Briefly introduce the biochemical background of galactose and only the most important physiological roles (3rd). This gives context to why it is critical to understand the findings from paragraph 3 in the context of those explained in paragraph 2 and vise-versa.

  4. Introduce the case of the beneficial effects of chronic oral D-galactose in the rat model of sporadic Alzheimer’s disease and main questions in the field that are indispensable for understanding the beneficial potential of galactose. Are the observed beneficial effects of D-galactose a consequence of its direct biochemical action in the brain or are they mediated by the effects in the gut? If the effects are consequences of its biochemical fate in the brain – this makes the case of factors regulating the exposure of tissues to D-galactose critical and the difference between parenteral (mostly harmful) and oral (mostly beneficial) administration might be a consequence of physiological buffering systems (e.g. controlled absorption, portal regulation,…). Considering D-galactose is able to induce secretion of some potentially protective molecules (e.g. GLP-1) from the gut it is also possible that the beneficial effects are of secondary nature. Finally, as some pathophysiological alterations of the gastrointestinal tract have been recently discovered in the STZ-icv rats (10.1159/000519988; 10.3390/antiox10071118) it is important to contextualize beneficial effects in this model and raise the question of whether we can consider the observed beneficial effects “translatable” to the controls as the protective effects might be mediated by e.g. restoring the physiological capacity of the intestine. All this is explained in paragraph 4, and while we understand that this makes the introduction quite long and rich with information, we are convinced that it is critical for the reader to be provided all the information to understand our aim and hypotheses. As Reviewer 2 requested the aim and hypotheses to be formulated more clearly, we rephrased the end of the Introduction section, and we hope that in its present form it is also easier to follow and more clear.

To summarize, we consider an extensive introduction helpful rather than superfluous in case of such a complex field with a lack of review articles that would objectively discuss the problem of “isolated research bubbles” and “selective exploration” of the effects of an omnipresent monosaccharide we all consume (and produce endogenously) daily. As we consider all the paragraphs indispensable to understanding the scope of the problem and the aim of the presented research, we are concerned that shortening the introduction might result in readers not grasping fully the problems in the field and the implications of the conclusions presented in the manuscript. We would therefore respectfully suggest that the introduction and the discussion that contextualize the findings remain in the “extensive” form to comply with what is considered to be good research practice and prevent forming oversimplified/unjustified conclusions.

We completely agree that “it is hard to understand in what conditions galactose acts as OS stimulant, and when it provides beneficial effects to cognitive functions”, however, we do not agree that this problem should be oversimplified as it has been in the Abstract (due to the lack of space) in a way that “chronic parenteral galactose administration triggers OS, and oral administration provides beneficial effects to the body”. This has been very clearly shown by Chogtu et al. who reported beneficial effects following both oral and subcutaneous administration of D-galactose on cognition of Wistar rats in the acute setting lost during chronic dosing (10.9758/cpn.2018.16.2.153), and Budni et al. who have shown that even oral administration of relatively low concentrations of D-galactose (100mg/kg) is able to induce cognitive impairment and oxidative stress when administered by repeated bolus dosing (10.1016/j.bbr.2015.12.041). We included this in the manuscript as well to prevent the formation of unjustified conclusions regarding the route of administration being the sole predictor of the harmful/beneficial effects of D-galactose.

Did the Authors obtain the approval of the appropriate ethic comitee?

Yes, this is reported under “Ethics approval:” in the manuscript as follows:

“All experiments were conducted in concordance with the highest standard of animal welfare. Only certified personnel handled animals. Animal procedures were carried out at the University of Zagreb Medical School (Zagreb, Croatia) and complied with current institutional, national (The Animal Protection Act, NN135/2006; NN 47/2011), and international (Directive 2010/63/EU) guidelines governing the use of experimental animals. The experiments were approved by the national regulatory body responsible for issuing ethical approvals, the Croatian Ministry of Agriculture (EP 186 /2018), and the Ethical Committee of the University of Zagreb School of Medicine (380-59-10106-18-111/173).”

The Authors wrote that studies on the role of galactose in animals are bias and their scientific value may be low. But, going through the manuscript I do not see any information how many repetitions of each measurments were performed. How can we be sure that these results are reliable and reproducible?

Thank you for this question. As emphasized in the manuscript, the question of susceptibility to bias has been brought up by others in the field as well (e.g. 10.1371/journal.pone.0184122) and we strongly believe measures to reduce the risk of bias should be implemented whenever possible and to the largest possible extent. In this manuscript and in the reported experiment we tried to maximally reduce the risk of bias using the following approach:

  1. Due to the experimental design, we were not able to use blinding (reported) and we assigned animals to groups using stratified randomization to control for possible effects of body mass and housing. All animals in the experiment were of the same age and underwent treatment at the same time, were exposed to the same food pellets, drinking water, and cage bedding.
  2. We analyzed “acute time response” rather than only a single time point to generalize acute effects in the gut (this provided more information on changes that demonstrated well-defined temporal patterns and increased our confidence in some of the observed changes – particularly those well-defined by monotonic relationships). Consistent changes that demonstrated no obvious temporal alterations were additionally analyzed by “overall effects” models (e.g. Fig 2C) so in this case, power to detect possible alterations was maintained at the highest level possible given the experimental design without undermining the biological soundness.

  3. We analyzed more than one part of the small intestine as we hypothesized the effects might be affected by anatomical/physiological differences of the cells in different parts of the intestine and by total exposure to galactose that changes along the gastrointestinal tract following oral administration.

  4. We did not analyze a single “marker of oxidative stress“ as is often the case. In contrast, we decided to focus on several mutually dependent enzymes and substrates contributing to the overall redox homeostasis. This enabled us to better understand the biological context of the observed alterations and increased our confidence in the observed alterations (e.g. consistent changes in biologically interdependent substrates of the nucleophilic arm of redox homeostasis – NADPH and LMWT in both the duodenum and ileum). This has been emphasized and elaborated in the manuscript.

  5. All the results were analyzed by statistical modeling in a way that enabled us to correct for important co-variates (e.g. all the measurements were corrected for protein concentration, but some were also corrected for other important biological covariates e.g. NADPH for NADP, H2O2 dissociation rate for the noise that might have been introduced by variable baseline absorbance due to baseline H2O2 and/or sample-induced biochemical interference).

  6. The results were analyzed and reported by “censoring” temporal information to provide the most reliable results at the expense of “power” to communicate the observed findings carefully, without excessive optimism and with a dose of additional skepticism. This “conservative” approach has been explained and elaborated in the manuscript (especially in the “Data analysis” section).

  7. We conducted a multivariate analysis of all parameters in both anatomical locations to uncover the variables that clustered together and that might represent the same biological phenomenon and/or be important to consider/model together due to interdependence.

  8. As there were just 6 animals per group and as it was not possible to assess sample size a priori as the primary outcome was a “composite” of different oxidative stress-related parameters (emphasized and explained in the manuscript) the experiment was most likely underpowered for specific redox parameters due to biological variation and potential inherent methodological variability. For this reason, all the results were reported as effects plots (as recommended by the current guidelines on reporting animal research (ARRIVE 2.0)) to communicate effects even if (due to methodological/biological variation) the differences between the groups did not meet conservative criteria for “statistically significant difference” predetermined by setting the α to 5% (emphasized in the “Data analysis” section).

From the “Explanation and elaboration for the ARRIVE guidelines 2.0” (10.1371/journal.pbio.3000411)

10b. If applicable, the effect size with a confidence interval.

Explanation. In hypothesis-testing studies using inferential statistics, investigators frequently confuse statistical significance and small p-values with biological or clinical importance [149]. Statistical significance is usually quantified and evaluated against a preassigned threshold, with p < 0.05 often used as a convention. However, statistical significance is heavily influenced by sample size and variation in the data (see Item 2. Sample size). Investigators must consider the size of the effect that was observed and whether this is a biologically relevant change.

Effect sizes are often not reported in animal research, but they are relevant to both exploratory and hypothesis-testing studies. An effect size is a quantitative measure that estimates the magnitude of differences between groups or strength of relationships between variables. It can be used to assess the patterns in the data collected and make inferences about the wider population from which the sample came. The confidence interval for the effect indicates how precisely the effect has been estimated and tells the reader about the strength of the effect [150]. In studies in which statistical power is low and/or hypothesis-testing is inappropriate, providing the effect size and confidence interval indicates how small or large an effect might really be, so a reader can judge the biological significance of the data [151,152]. Reporting effect sizes with confidence intervals also facilitates extraction of useful data for systematic review and meta-analysis. When multiple independent studies included in a meta-analysis show quantitatively similar effects, even if each is statistically nonsignificant, this provides powerful evidence that a relationship is ‘real’, although small.

Report all analyses performed, even those providing non-statistically significant results. Report the effect size to indicate the size of the difference between groups in the study, with a confidence interval to indicate the precision of the effect size estimate.

Regarding the “repetitions of measurements”, in all the biochemical methods used we used estimates obtained from technical replicates at various levels wherever possible (e.g. for all the measurements made using plate reader absorbance estimate was based on at least 25 measurements of each sample). When measuring each parameter, we first ensured the measurements were in the quantitative range for each specific variable and then repeated the measurements using appropriate adjustments. E.g. for MDA measurements, usually we first do the test run to assess the appropriate volume of tissue that has to be used in the experiment to extract the amount of analyte that falls within the linear range of the model derived from standard samples using the same reaction conditions (e.g. heating time/temperature used to speed up the reaction). In the test reactions, we determine the optimal volume of tissue, heating temperature and time, and the volume of solvent used for the extraction of the TBA-MDA adduct. As this is a common procedure standardly used for all biochemical measurements to ensure they truly make sense for the samples to be analyzed we didn’t describe the standard validation procedures in detail in the materials and methods section.

As many analyses were done, and a substantial volume of samples had to be spent on validation procedures, we couldn’t afford additional technical replicates of each sample just to control for potential sampling bias. Instead, standard measures were used to reduce the risk of faulty “sampling” of each sample by pipette (e.g. all the samples were thoroughly vortexed each time prior to pipetting), and the variation that may have been introduced by different steps of each biochemical method used in the experiment has been previously assessed using standard procedures of bioanalytical method validation (e.g.  10.1080/10715762.2021.1912340 [manuscript and supplement]; 10.1101/2020.06.16.154682 [manuscript and supplement]).

All the abovementioned measures have been employed to maximally reduce the risk of bias in the reported experiment/manuscript, and transparently communicate the observed changes of redox homeostasis upon oral D-galactose administration in the rat small intestine. We thank Reviewer 1 for this question, and - because this is open peer-review that will be available to the reader alongside the manuscript - we believe our answer will help the reader take into account what we have done to reduce the risk of bias, and what could not have been done due to time constraints, biological (limited volume of samples) and financial limitations, and most importantly to interpret the reported results in this context.

line 633 - "The intestine is the primary organ for food digestion, absorption, and metabolism". Rather, the first organ involved in digestion is the stomach, not the intestine.

Thank you for pointing this out. We didn’t want to state that the intestine is the first organ involved in food digestion anatomically as the digestion process obviously starts in the mouth. Instead, we wanted to emphasize the intestine is important for all the functions that have been enumerated. The sentence has now been altered so it is clear to the reader what we intended to say.

In general, both parts Results and also Discussion, are too extensive, and it is difficult to understand what the research actually shows. Please make it more concise and provide only the most important information resulting from the conducted research. 

Thank you for this suggestion. As elaborated in the answer to the first comment, and in line with the approach used to reduce the risk of bias, we didn’t want to oversimplify the results and encourage the formation of unfounded conclusions based on a rather small exploratory experiment. For this reason, all the findings were described in the results sections and discussed (together with alternative hypotheses) in the broader context of what they may mean biologically in the discussion. Providing only “the most important information” promotes misinterpretation of the results and the formation of unjustified conclusions based on the establishment of simple theoretical models that are usually ill-equipped to provide reliable estimates of the reality. The latter has been widely debated in the (bio)medical community and is best reflected in the age-old discussions on how to communicate (complete and un-trimmed) scientific evidence (very nicely covered in the recent “Five Rules for Evidence Communication” by Blastland et al.), and fruitful discussions on how the preoccupation of biomedical scientists with prioritizing one piece of evidence over the other based on subjective criteria often results in the formation of unjustified conclusions (this is very nicely reflected in the common cognitive bias that provides the foundations for the unfounded idea that “statistical significance” is reflective of biological significance). The problem of “selective reporting”, based on what one considers “the most important” has been discussed in the clinical and preclinical context (10.1136/bmj.c869; 10.1111/j.1469-185X.2007.00027.x), and the official guidelines for reporting the results from animal experiments strongly support providing effects sizes and confidence intervals and reporting all analyses so “a reader can judge the biological significance of the data” (ARRIVE 2.0) regardless of what may seems at the cursory look as “the most important” usually based on conservative determination of “statistical significance”. We would therefore respectfully suggest that the introduction and the discussion that contextualize the findings remain in the “extensive” form (as neither MDPI nor Antioxidant limit the length of the manuscript in any way for this purpose) to comply with what is considered to be good research practice and prevent forming oversimplified/unjustified conclusions.

Thank you very much for the serious and professional peer review and constructive criticism provided. Thank you very much for allocating time for providing a review for our manuscript!

Sincerely,

Jan Homolak MD (for the authors)

Reviewer 2 Report

The manuscript “The effect of acute oral galactose administration on the redox system of the rat small intestine” by Homolak explore the impact of galactose on the redox activities taking place at the  small intestine. The study is interesting and robust experimental design and well-presented and discussed results. The manuscript is publishable subject to minor changes:

  • Authors must elaborate more on previous relevant studies about the potential effect of oral galactose administration on the redox system – the introduction is not very easy to follow/read.
  • Authors also must elaborate more on their aim and hypothesis, as it was very briefly mentioned by the end of their introduction.
  • How did the author confirm that the numbers of subjects used (each group, no= 6) are statistically valid?
  • What are the implications of this study in human?

Author Response

A point-by-point response to reviewers regarding the manuscript: „ The effect of acute oral galactose administration on the redox system of the rat small intestine“

Reviewer 2

We wish to thank Reviewer 2 for their time and effort in reviewing our manuscript. We strongly believe addressing the comments improved the quality (and reliability) of the data we presented and the quality of the manuscript in general.

The manuscript “The effect of acute oral galactose administration on the redox system of the rat small intestine” by Homolak explore the impact of galactose on the redox activities taking place at the  small intestine. The study is interesting and robust experimental design and well-presented and discussed results. The manuscript is publishable subject to minor changes:

Authors must elaborate more on previous relevant studies about the potential effect of oral galactose administration on the redox system – the introduction is not very easy to follow/read.

Thank you for your suggestions. Unfortunately, studies exploring the effects of oral galactose administration on the redox system are scarce. In fact, the paucity of evidence on the effects of galactose on redox homeostasis following physiological (oral) administration was one of the main reasons we conducted the study presented in the manuscript and why we are currently conducting other studies to assess the effects of chronic oral D-galactose administration on redox homeostasis. To the best of our knowledge, the only evidence on the effects of oral D-galactose on redox homeostasis has been published by Josiane Budni et al. (10.1016/j.bbr.2015.12.041; 10.1007/s11011-017-9972-9) who reported the development of cognitive impairment, oxidative stress, and mitochondrial deficits following repeated bolus dosing of relatively low doses of D-galactose (100 mg/kg). We believe the work of Budni et al., and that of Chogtu et al. (10.9758/cpn.2018.16.2.153) who reported a time-dependent positive effect of D-galactose on learning and memory after both oral and subcutaneous administration suggests that the effects of D-galactose might be dependent on the buffering capacity of the body that has the largest capacity when D-galactose enters the body via the oral (physiological) route and has to pass the absorptive barrier of the intestine, the portal circulation and the largest galactose metabolizer in the body (the liver; - galactose has in fact previously been used to assess liver function in humans!). We have now also added the work of Budni and Chogtu in the introduction we have originally left out because we were aware the introduction in its present extensive form may be challenging to follow (please see our response to Reviewer 1).

We hope that in its present form, the introduction now mentions all the important information for the reader to understand both the rather confusing and paradoxical research setting pertaining to the effects of D-galactose, and the main pieces of evidence that led us to formulate the aim of our present research.

Authors also must elaborate more on their aim and hypothesis, as it was very briefly mentioned by the end of their introduction.

Thank you for your suggestion. We altered the introduction so it is more clear, and we restructured the way we present the aims of the presented research and the hypotheses. We hope that in its present form the introduction provides a clearer overview of what we intended to assess with this experiment.

How did the author confirm that the numbers of subjects used (each group, no= 6) are statistically valid?

Thank you for this question. As explained in the manuscript in the methods section under „Data analysis“ and in the „Limitations“ section, we were not able to assess sample size a priori due to the primary outcome (redox system) being a „composite“ of oxidative stress-related parameters measured with independent techniques that standardly have to be adapted to meet optimal sensitivity given available specimens (e.g. larger volumes of samples, longer extraction times, more flashes per well) on a case-by-case basis. Furthermore, although we did report the results based on group/time-point differences what makes the question of whether 6 is a large enough number of animals per group to detect certain effects absolutely valid, we have emphasized throughout the manuscript that trends over time were in our primary focus, and that we only used group comparisons as we considered this „conservative“ approach to be more transparent and informative (communicating large uncertainties arising from a small number of animals per group and independence of measurements in time), but not at the expense of appreciation of temporal trends of both monotonic and non-monotonic relationships given appropriate reporting and visualization. Having said this, the analytical approach we used for the analyses, described in the manuscript in detail, was used with the sole purpose of „overcoming “ the problem of whether n=6 is a large enough number of animals per group to detect the effects we were interested in.

The problem that most animal studies are underpowered to detect certain effects is very well known and it is usually due to i) ethical reasons (e.g. experiments with 30 animals/group are not very common); ii) logistical reasons (it is not very practical to have so many animals/group that methodological and logistical reasons introduce greater noise than what is present due to biological variability – e.g. a large number of animals means that all cannot be euthanized at the same time… maybe some will be euthanized in the morning and others in the evening introducing a substantial amount of noise due to diurnal variability of many biological functions…); iii) the problem that power is usually calculated for the primary outcome, and in animal research, there are usually many outcomes of interest that can (and should) be determined with variable levels of „power to detect“. The repercussion of the latter is the fact that researchers often misinterpret „statistical significance“ as an indication of biological significance, and many large effects that may be biologically significant remain unnoticed because, although the effects were large, they didn't meet the criteria for conservative „statistical significance“ due to large variance reflecting methodological or biological variability.

Consequently, the problem of power, the Reviewer 2 refers to as „statistical validity“ is common in animal research.

The way it is supposed to be dealt with is that the primary outcome should be determined whenever possible, and power should be assessed to calculate the number of animals needed for the experiment (in this „optimal“ scenario, many other secondary effects of interest will often still be underpowered).

This approach is not particularly practical as it doesn't take into account that in biological research analytical methods are supposed to be adapted to answer specific questions (so it is not clear how one should exactly determine the performance of the method a priori) and it doesn't provide an answer to what should be done with all the other „underpowered“ analyses. The way official guidelines for the reporting of in vivo experiments (ARRIVE) recommend one should tackle this problem is to report all the analyses with effect size estimates (determining the size of the effect) with confidence intervals (communicating the reliability of estimates and the precision with which the effect size has been estimated).

From the “Explanation and elaboration for the ARRIVE guidelines 2.0” (10.1371/journal.pbio.3000411)

10b. If applicable, the effect size with a confidence interval.

Explanation. In hypothesis-testing studies using inferential statistics, investigators frequently confuse statistical significance and small p-values with biological or clinical importance [149]. Statistical significance is usually quantified and evaluated against a preassigned threshold, with p < 0.05 often used as a convention. However, statistical significance is heavily influenced by sample size and variation in the data (see Item 2. Sample size). Investigators must consider the size of the effect that was observed and whether this is a biologically relevant change.

Effect sizes are often not reported in animal research, but they are relevant to both exploratory and hypothesis-testing studies. An effect size is a quantitative measure that estimates the magnitude of differences between groups or strength of relationships between variables. It can be used to assess the patterns in the data collected and make inferences about the wider population from which the sample came. The confidence interval for the effect indicates how precisely the effect has been estimated and tells the reader about the strength of the effect [150]. In studies in which statistical power is low and/or hypothesis-testing is inappropriate, providing the effect size and confidence interval indicates how small or large an effect might really be, so a reader can judge the biological significance of the data [151,152]. Reporting effect sizes with confidence intervals also facilitates extraction of useful data for systematic review and meta-analysis. When multiple independent studies included in a meta-analysis show quantitatively similar effects, even if each is statistically nonsignificant, this provides powerful evidence that a relationship is ‘real’, although small.

Report all analyses performed, even those providing non-statistically significant results. Report the effect size to indicate the size of the difference between groups in the study, with a confidence interval to indicate the precision of the effect size estimate.

To overcome this problem, and to make sure i) we communicate our results with maximal transparency, and ii) that we don't make the mistake of interpreting statistical significance as biological significance – we reported all our results strictly following the ARRIVE 2.0 guidelines reporting model estimates of all variables alongside contrasts visualized using effects plots to communicate effect sizes with accompanying uncertainties depicted by confidence intervals of estimates. This way, as elaborated (and emphasized) in the methods section „Data analysis“ and in the „Limitations“ section, we reported all the observed biological effects regardless of whether the group size and variance (resulting from either inherent biological variability or analytical steps) introduced the noise that „pushed“ some effects above the pre-assigned threshold of „statistical significance“ (α was set at 5%). So, regardless of whether n=6 was or wasn't enough to observe particular effects with large enough certainty, we reported all the important information for the reader to „judge the biological significance of the data“ as suggested by ARRIVE 2.0. This way we provided critical information for the potential subsequent meta-analyses of studies that may aim to provide the answer to the question of the effects of acute oral D-galactose on redox homeostasis in the rat intestine that may be too small to be reliably assessed in this n=6/group experiment, and we offer information that may be important for defining (Bayesian) priors for groups that may aim to assess the similar problem with additional animal experiments (please see Small Sample Size Solutions: A Guide for Applied Researchers and Practitioners; ISBN 9780367222222)

What are the implications of this study in human?

The presented results cannot be directly translated to humans. Nevertheless, as this study provides the foundations for understanding the (neuro)protective effects of D-galactose in rats, elucidating the mechanisms responsible for the observed effects may help us understand whether oral administration of small doses of D-galactose may also be able to exert neuroprotective effects in humans. Considering there's a great need for novel neuroprotective strategies in neurodegeneration and that at the current moment no successful disease-modifying therapies exist, we believe this to be of great importance. Furthermore, humans ingest a substantial amount of galactose daily and produce it endogenously, and the question of whether this sugar (mostly considered harmful in the literature) actually contributes to the development of diseases in humans has never been seriously considered. The presented results provide preliminary evidence that (at least when ingested via the physiological route (oral), and in the acute setting) D-galactose is not able to induce oxidative stress and „cause harm“ in rats. Although this cannot be translated to humans, if additional studies show that this is also the case following chronic administration, we consider this to be reassuring evidence for the formation of the hypothesis that D-galactose may be less harmful than we think if administered via the oral route in mammals. Of course, it goes without saying that this provides only very weak hypothesis-generating evidence for humans and whether this really is the case remains to be tested in human subjects in the future.

Thank you very much for the serious and professional peer review and constructive criticism provided. Thank you very much for allocating time for providing a review for our manuscript!

Sincerely,

Jan Homolak MD (for the authors)

Round 2

Reviewer 1 Report

The Authors did not address the majority of my questions, instead of that presented a large elaborate trying to explain why they do not want do implement changes which I have suggested.

I still think the manuscript is to large. I do believe that shortening the article will not reduce its quality and will not prevent the reader from being properly introduced to the topic.

Moreover I asked how many samples were analyzed and I also did not obtain a clear response. It should be clearly indicated how many samples were taken from each animal (ex. blood or other tissues).

The discussion part is also to expand. It may be shortened without losing any quality.

Author Response

A point-by-point response to reviewers regarding the manuscript: „ The effect of acute oral galactose administration on the redox system of the rat small intestine“

Reviewer 1

We wish to thank Reviewer 1 for their time and effort in reviewing our manuscript again. We strongly support the discussion and we believe it has the potential to improve the quality of the presented manuscript.

Comments and Suggestions for Authors

The Authors did not address the majority of my questions, instead of that presented a large elaborate trying to explain why they do not want do implement changes which I have suggested.

Thank you for your response. We take peer review seriously and with respect as we consider it fundamental for the structure of science. Our intention was to approach peer review with maximal respect considering all the suggestions from all the reviewers (some of which were diametrical – e.g. those pertaining to the introduction section). Furthermore, we tried to approach the peer review with respect to what it is and what it is not intended to address. We strongly believe the concept of scientific peer review is to discuss important conceptual and methodological issues before the manuscript can be officially published to prevent the formation of unjustified conclusions based on flawed methodology. We also strongly believe that peer review should not impose limitations that are under the jurisdiction of the publisher and that are not related to scientific soundness – e.g. the comments/suggestions/requirements pertaining to the form rather than the content such as the number of words or the structure of the text.

Having said this, we consider that we did address the majority of the questions that were raised. As stated, we wrote an extensive elaboration of why we do not agree with the opinion that the introduction is “too long”. We would like to believe that peer review is not supposed to be the unelaborated blind implementation of the “suggestions”/requirements just to please the reviewers and pass the peer review process, but a constructive, inclusive, and fruitful (scientific) discussion. Along this line, we provided a very clear elaboration of why we think that only the key concepts critical for the reader to understand the “general idea” of the work were presented in the introduction, and what was the motivation for the structure of paragraphs and their arrangement. We explained that we are concerned that shortening the introduction might result in the incomplete presentation of the complex problem and that this might encourage the reader to form unjustified and oversimplified conclusions. The latter was clearly illustrated by the suggestion: “It must be clearly stated (like in the Abstract), that chronic parenteral galactose administration triggers OS, and oral administration provides beneficial effects to the body.” to which we answered that we respectfully believe the statement is an oversimplification of the problem and provided scientific evidence to back up this claim.

We did not write the elaboration instead of restructuring the introduction because we are not open to suggestions, but because we believe that the text in its present form is well balanced in its complexity in regard to the complexity of the scientific problem. Although restructuring the text would have been easier for us if we just wanted to “pass the peer review”, we wanted to seriously discuss the problem, so we wrote a 1000-word answer to provide our perspective with the hope that fruitful and constructive discussion would arise that would help us understand how to rewrite the introduction (should we completely abandon general introduction paragraph at the beginning? Should we skip the overview of how and why galactose is used in rodent research on aging and why the literature on the galactose-induced aging model may provide only partial “black-and-white” insight into the effects of galactose? Should we skip the overview of its physiological functions that are in contrast with what the literature on the galactose-induced aging model proposes? Should we completely skip the part on the “paradoxical” results from the STZ-icv model that motivated us to do the research we did? Or should we try to condense all of the abovementioned parts of the introduction – and if yes with what purpose(?) as the text is already rich with information and the reader would more likely benefit from a “story” that helps them understand “the leading thought” than simply a fewer number of words in the introduction). Although we consider that the presentation of the problem and the form and length of the introduction is not something that should be in the focus of the peer review because it is obviously affected by subjective preferences (and not objective scientific criteria), we are still open for discussion on how the introduction should be restructured. We do not consider the suggestion that “it should be shortened” accompanied with the suggestion that the problem of the route of administration should be (incorrectly) oversimplified (parenteral=OS; peroral=beneficial) as valid and objective guidelines that we can try to fulfill to make the introduction shorter and easier to follow without undermining its scientific integrity.

The problem of subjective preferences is also reflected in the response of Reviewer 2 who suggested slight extension of the introduction and who rated the introduction with the highest mark in respect to the question “Does the introduction provide sufficient background and include all relevant references?” Do you have any suggestions on how we should approach the problem of shortening the introduction in regard to Reviewer 2 and whether we should again expand it after we shorten it due to opposing subjective preferences of reviewers?

Once again, we are willing to consider the restructuring of the introduction, but we require open and elaborated discussion and constructive (targeted) comments to be able to comprehend and approach the problem (although we would rather focus on the science).

I still think the manuscript is to large. I do believe that shortening the article will not reduce its quality and will not prevent the reader from being properly introduced to the topic.

We appreciate your opinion. Although we do not consider subjective comments pertaining to the form rather than the content should be in the focus, we are willing to reconsider the structure should specific constructive comments that would enable us to see the problem be provided. Please see the response to the first comment.

Moreover I asked how many samples were analyzed and I also did not obtain a clear response. It should be clearly indicated how many samples were taken from each animal (ex. blood or other tissues).

Thank you very much for pointing this out, this is an important problem that we have missed. In the schematic overview of the study, we stated the number of samples with respect to the duodenum, and not the number of animals in general (we corrected the schematic representation). This was a mistake as some samples were unfortunately stored at inadequate temperature (4/48) so they had to be excluded from the biochemical analyses and not the same samples were excluded in the duodenum and in the ileum. We have now clearly stated this in the methods section and emphasized that the mishandled samples did not undergo biochemical analyses out of precaution (they were not excluded post hoc). The samples that had to be excluded before the analyses were duodenum 30_3 and 120_6 and ileum 0_1 and 60_2. The number of samples analyzed in the duodenum (n0=6; n30=5; n60=6; n120=5) and in the ileum (n0=5; n30=6; n60=5; n120=6) is now clearly stated in the manuscript (Methods; Tissue collection and sample preparation) and in the schematic overview in Fig 1.

We have misinterpreted the original question of Reviewer 1 („The Authors wrote that studies on the role of galactose in animals are bias and their scientific value may be low. But, going through the manuscript I do not see any information how many repetitions of each measurments were performed. How can we be sure that these results are reliable and reproducible?“) and we were under the impression that we were asked about technical replicates, the reliability, and the reproducibility of the study so we provided the answer that described how we ensured the results were reliable and reproducible. We are sorry for the misunderstanding and we want to once again express gratitude for drawing our attention to this.

The discussion part is also to expand. It may be shortened without losing any quality.

Thank you for your suggestion. As stated in responses to previous comments, we do not consider there are unscientific/unsupported claims in the discussion and if there are please suggest the problematic ones and we will be more than pleased to correct them. Regarding the subjective preferences pertaining to the length of the text, we stand by our opinion and we are willing to consider restructuring the text if specific changes are suggested as we believe all parts are valuable for the discussion of the observed changes and will thus benefit the reader. We are not concerned that the reader will „become bored“ with the scientific article if the text is „too long“ as this is not a newspaper or a popular science article. On contrary, all information and context can only help one understand the problem from a broader perspective and thus guard against forming unjustified/oversimplified conclusions. Luckily, Antioxidants is an online-only scientific journal so there are no material or environmental policy-induced constraints pertaining to the length of the manuscript so there should be no word limits that come at the expense of scientific quality („Antioxidants has no restrictions on the length of manuscripts, provided that the text is concise and comprehensive“). We understand that the text may not be concise, but given the problem we discuss, we believe a more concise text would be less comprehensive.

Thank you for your comments and suggestions and for allocating time to contribute to the quality of the manuscript.

Sincerely,

Jan Homolak, MD (for the authors)